# ONLINE REINFORCEMENT LEARNING IN NON-STATIONARY CONTEXT-DRIVEN ENVIRONMENTS

**Pouya Hamadanian**
MIT CSAIL
pouyah@mit.edu

**Arash Nasr-Esfahany**
MIT CSAIL
arashne@mit.edu

**Malte Schwarzkopf**
CS Brown University
malte@cs.brown.edu

**Siddhartha Sen**
Microsoft Research
sidsen@microsoft.com

**Mohammad Alizadeh**
MIT CSAIL
alizadeh@mit.edu

## ABSTRACT

We study online reinforcement learning (RL) in non-stationary environments, where a time-varying exogenous context process affects the environment dynamics. Online RL is challenging in such environments due to "catastrophic forgetting" (CF). The agent tends to forget prior knowledge as it trains on new experiences. Prior approaches to mitigate this issue assume task labels (which are often not available in practice), employ brittle regularization heuristics, or use off-policy methods that suffer from instability and poor performance.

We present Locally Constrained Policy Optimization (LCPO), an online RL approach that combats CF by anchoring policy outputs on old experiences while optimizing the return on current experiences. To perform this anchoring, LCPO locally constrains policy optimization using samples from experiences that lie outside of the current context distribution. We evaluate LCPO in Mujoco, classic control and computer systems environments with a variety of synthetic and real context traces, and find that it outperforms a variety of baselines in the non-stationary setting, while achieving results on-par with a "prescient" agent trained offline across all context traces.

LCPO's source code is available at https://github.com/pouyahmdn/LCPO.

## 1 INTRODUCTION

*— Those who cannot remember the past are condemned to repeat it. (George Santayana, The Life of Reason, 1905)*

Reinforcement Learning (RL) has seen success in many domains (Mao et al., 2017; Haarnoja et al., 2018a; Mao et al., 2019b; Marcus et al., 2019; Zhu et al., 2020; Haydari & Yılmaz, 2022), but real-world deployments have been rare. A major hurdle has been the gap between simulation and reality, where the environment simulators do not match the real-world dynamics. Thus, recent work has turned to applying RL in an *online* fashion, i.e., continuously training and using an agent in a live environment (Zhang et al., 2021; Gu et al., 2021).

While online RL is difficult in and of itself, it is particularly challenging in *non-stationary* environments— also known as continual RL (Khetarpal et al., 2020)—where the characteristics of the environment change over time. A key challenge is Catastrophic Forgetting (CF) (McCloskey & Cohen, 1989). An agent based on function approximators like Neural Networks (NNs) tends to forget its past knowledge when training sequentially on new non-stationary data. On-policy RL algorithms (Sutton & Barto, 2018) are particularly vulnerable to CF in non-stationary environments, since these methods cannot retrain on stale data from prior experiences.

In this paper, we consider problems where the source of the non-stationarity is an observed exogenous *context* process that varies over time and exposes the agent to different environment dynamics. Such context-driven environments (Sinclair et al., 2023; Mao et al., 2018; Zhang et al., 2023; Dietterich et al., 2018; Pan et al., 2022) appear in a variety of applications. Examples include computer systems subject to incoming workloads (Mao et al., 2018), locomotion in environments with varying terrains and

obstacles (Heess et al., 2017b), robots subject to external forces (Pinto et al., 2017), and more. In contrast to most prior work (Alegre et al., 2021; Chandak et al., 2020), we do not restrict the context process to be discrete, piecewise stationary or Markov.

Broadly speaking, there are three existing approaches to mitigate CF in online learning. One class of techniques is *task-based* (Rusu et al., 2016; Kirkpatrick et al., 2017; Schwarz et al., 2018; Farajtabar et al., 2019; Zeng et al., 2019). Such works assume explicit task labels that identify the different context distributions which the agent encounters over time. Task labels make it easier to prevent the training for one context from affecting knowledge learned for other contexts. In settings where task labels (or boundaries) are not available, a few proposals try to infer the task labels via self-supervised (Nagabandi et al., 2019b) or Change-Point Detection (CPD) approaches (Padakandla et al., 2020; Alegre et al., 2021). These techniques, however, are brittle when the context processes are difficult to separate and task boundaries are not pronounced (Hamadanian et al., 2022). Our experiments show that erroneous task labels lead to poorly performing agents in such environments (§5).

A second category of approaches avoids task labels by approximating task-based methods with heuristics (Schwarz et al., 2018; Chaudhry et al., 2018; Kaplanis et al., 2018; Woo et al., 2022). However, these heuristics are based on brittle assumptions about the nature and cadence of non-stationarity. For example, one approach implicitly assumes each episode is a distinct task, and uses a window of past $N$ episode to regularize learning (Woo et al., 2022). These assumptions are rarely met and would likely lead to poor performance in practice, as we observe in our analysis and evaluations (§5 and §D.1, §D.2 and §D.3 in the Appendix).

The third category of approaches employs rehearsal, i.e., learning using past or generated data. For example, off-policy learning (Sutton & Barto, 2018) makes it possible to retrain on past data. These techniques (e.g., Experience Replay (Mnih et al., 2013), CLEAR (Rolnick et al., 2019), etc.) store prior experience data in a buffer and sample from the buffer randomly to train. Not only does this improve sample complexity, it sidesteps the pitfalls of sequential learning and prevents CF (Rolnick et al., 2019). However, off-policy methods come at the cost of increased hyper-parameter sensitivity and unstable training (Duan et al., 2016; Gu et al., 2016; Haarnoja et al., 2018b). This brittleness is particularly catastrophic in an online setting, as we also observe in our experiments (§5).

We present LCPO (§4.1), an on-policy RL algorithm that "anchors" policy outputs for old contexts while optimizing for the current context. Unlike prior work, LCPO does not rely on task labels and only requires an Out-of-Distribution (OOD) detector, i.e., a function that recognizes old experiences that occurred in a sufficiently different context than the current one. LCPO maintains a bounded buffer of past experiences, similar to off-policy methods (Mnih et al., 2013). But as an on-policy approach, LCPO does not use stale experiences to optimize the policy. Instead, it uses past data to *constrain* the policy optimization on fresh data, such that the agent's behavior does not change in other contexts.

We evaluate LCPO on several environments with real and synthetic contexts (§5), and show that it outperforms a variety of baselines across mentioned categories in the online learning setting. We also compare against a "prescient agent" that is trained offline on the entire context distribution prior to deployment. The prescient agent does not suffer from CF. Among all the online methods, LCPO is the closest to this idealized baseline. Our ablation results show that LCPO is robust to variations in the OOD detector's thresholds and works well with small experience buffer sizes.

LCPO's source code is available online at `https://github.com/pouyahmdn/LCPO`.

## 2  PRELIMINARIES

**Notation.**  We consider online reinforcement learning in a non-stationary context-driven Markov Decision Process (MDP), where the context is observed (only up to the current time step $t$) and exogenous. Formally, at time step $t$ the environment has state $s_t \in \mathcal{S}$ and context $z_t \in \mathcal{Z}$. The agent takes action $a_t \in \mathcal{A}$ based on the observed state $s_t$ and context $z_t$, $a_t = \pi(s_t, z_t)$, and receives feedback in the form of a scalar reward $r_t = r(s_t, z_t, a_t)$, where $r(\cdot, \cdot, \cdot) : \mathcal{S} \times \mathcal{Z} \times \mathcal{A} \to \mathbb{R}$ is the reward function. The environment's state, the context, and the agent's action determine the next state, $s_{t+1}$, according to a transition kernel, $T(s_{t+1}|s_t, z_t, a_t)$. The context $z_t$ is an independent stochastic process, unaffected by states $s_t$ or actions $a_t$. Finally, $d_0$ defines the distribution over initial states ($s_0$). This model is fully defined by the tuple $\mathcal{M} = (\mathcal{S}, \mathcal{Z}, \mathcal{A}, \{z_t\}_{t=1}^{\infty}, T, d_0, r)$.

**Non-stationary contexts.** The non-stationary context $\mathbf{z} = \{z_t\}_{t=1}^{\infty}$ impacts the environment dynamics and implies a non-stationary environment. We assume the context process can change arbitrarily: e.g., it can follow a predictable pattern, be i.i.d samples from some distribution, be a discrete process or a multi-dimensional continuous process, experience smooth or dramatic shifts, be piecewise stationary, or include any mixture of the above. We have no prior knowledge of the context process, the environment dynamics, or access to an offline simulator. Examples of (observed) context processes include market demand in a supply chain system, incoming request workloads in virtual machine allocation problem, customer distributions in airline revenue management (Sinclair et al., 2023), traffic information in vehicular networks (Wu et al., 2017), terrain profiles in a locomotion task (Heess et al., 2017a), network traffic for video streaming (Mao et al., 2020) and congestion control (Winstein & Balakrishnan, 2013), etc.

**Goal.** We seek good long-term performance. Formally, for a given policy $\pi : \mathcal{S} \times \mathcal{Z} \rightarrow \mathcal{A}$ and context process $\mathbf{z} = \{z_t\}_{t=1}^{\infty}$ we define the lifelong return as $J(\pi, \mathbf{z}) = \lim_{t \to \infty} \sum_{i=1}^{t} \frac{r_i}{t}$ for an infinite horizon MDP. Similarly, for finite horizon MDPs of length $H$, where episode $i$ is subject to context traces $\mathbf{z}_i = (z_{H.i}, z_{H.i+1}, ..., z_{H.(i+1)-1})$ and has an episodic return of $R_i = \sum_{t=1}^{H} r_t^{(i)}$, we define the lifelong return as $J(\pi, \mathbf{z}) = \lim_{t \to \infty} \sum_{i=1}^{t} \frac{R_i}{t}$. For a policy sequence $\mathbf{\Pi} = \{\pi_t\}_{t=1}^{\infty}$, e.g., the sequence of policies resulting from a continual RL algorithm, we can define the lifelong return $J(\mathbf{\Pi}, \mathbf{z})$ similarly.

**Online RL.** In most RL settings, a policy is trained in a separate training phase. During testing, the policy is fixed and does not change. By contrast, online learning starts with the test phase, and the policy must reach and maintain optimality within this test phase. An important constraint in the online setting is that the agent gets to experience each interaction only once. There is no way to revisit past interactions and try a different action in the same context. This is a key distinction with training in a separate offline phase, such as in meta-learning (Al-Shedivat et al., 2018), where the agent can explore the same conditions many times.

Note that the policy that maximizes lifelong return $\pi^* = arg\max_{\pi} J(\pi, \mathbf{z})$ has to be *prescient*, i.e., it needs to have upfront knowledge of the context process $\mathbf{z}$. Since an online agent is causal and has only observed context values up to the current time step $t$, it can never perform as well as this prescient policy. Therefore, in general online RL agents will have a gap with prescient policies in terms of lifelong return. In certain special cases the online RL can asymptotically reach the prescient policy, e.g., when the context process is Markovian the entire context-driven MDP collapses to a standard MDP with a state $\tilde{s}_t = <s_t, z_t>$. However, we do not intend to limit the context process in any way, and our aim it to minimize the gap between the online and prescient agents for arbitrary context processes.

## 3 RELATED WORK

**Non-stationary RL.** Non-stationary RL is a family of sub-problems, such as CF, latent context inference, meta-learning, etc (Khetarpal et al., 2020). In this work we focus on CF, and highlight the differences of CF with other well-known non-stationary RL problems below. Then, we will explore related work for CF in Machine Learning (ML) and RL. We highlight other lines of work in §A in the Appendix.

**Latent Context Inference.** These works consider a context-driven MDP where the context $z_t$ is unobserved. The goal is to infer an estimated context $\hat{z}_t$ from other signals, such as transition functions, reward functions, etc (Hallak et al., 2015; Zintgraf et al., 2019; Xie et al., 2020; Caccia et al., 2020; Lee et al., 2020; He et al., 2020; Poiani et al., 2021; Chen et al., 2022; Huang et al., 2022; Feng et al., 2022; Ren et al., 2022; Woo et al., 2022; Bing et al., 2022; Luo et al., 2022; Lee et al., 2023). Once inferred, a traditional RL algorithm such as Soft Actor Critic (SAC) learns a policy $\pi(\cdot|s_t, \hat{z}_t)$ from the state and inferred context, and is typically compared to an 'upper-bound policy' that observes the true context $\pi(\cdot|s_t, z_t)$. These works aim to recover the unobserved context, while we focus on CF after observing the true/recovered context. In fact, the 'upper-bound' policies in these works are baselines we compare to in §5. Combining LCPO with this line of work to solve CF in environments with latent context is an interesting future work.

**Catastrophic Forgetting.** Three general techniques exist for mitigating CF in ML (Parisi et al., 2019); (1) regularizing the optimization to avoid memory loss during sequential training (Kirkpatrick et al., 2017; Zenke et al., 2017; Farajtabar et al., 2019; Lopez-Paz & Ranzato, 2022); (2) training separate parameters per task, and expanding/shrinking parameters as necessary (Rusu et al., 2016; Shmelkov et al., 2017; Li et al., 2019); (3) rehearsal, i.e., retraining on original data or generative batches (Shin et al., 2017; Isele & Cosgun, 2018; Atkinson et al., 2021); or combinations of these techniques (Schwarz et al., 2018; Aljundi et al., 2019).

Regularization techniques such as Elastic Weight Consolidation (EWC) and Orthogonal Gradient Descent (OGD) require task labels. Approximations have been proposed for problems without task labels (Schwarz et al., 2018) or boundaries (Woo et al., 2022). Kaplanis et al. (2018) use biologically inspired models of brain synapses to regularize networks.

Another class of approaches aims to infer task labels, by learning the transition dynamics of the MDP, and detecting a new environment when a surprising sample is observed with respect to the learned model (Doya et al., 2002; da Silva et al., 2006; Padakandla et al., 2020; Alegre et al., 2021). The inferred labels are often used to train separate policies/models to mitigate CF. These methods are effective when MDP transitions are abrupt and have well-defined boundaries, but are brittle and perform poorly in realistic environments with noisy and hard-to-distinguish non-stationarities (Hamadanian et al., 2022).

For rehearsal, we can use learned models of the MDP to replay past experiences (Xu et al., 2020; Lee et al., 2020; Pong et al., 2020; Huang et al., 2021; Janner et al., 2021). An alternative type of rehearsal is off-policy training (Haarnoja et al., 2018b; van Hasselt et al., 2016) (e.g., Experience Replay (Mnih et al., 2013)), which can train on stale data, naturally circumvent sequential learning and avoid CF. However, off-policy RL is empirically unstable and sensitive to hyperparameters due to bootstrapping and function approximation (Sutton & Barto, 2018), and is often outperformed by on-policy algorithms in online settings (Duan et al., 2016; Gu et al., 2016; Haarnoja et al., 2018b). CLEAR (Rolnick et al., 2019) is an off-policy RL algorithm explicitly designed to overcome CF with fast adaptations. Similarly, PT-DQN (Anand & Precup, 2023) learns a permanent Q-network to remember past tasks while learning a transient Q-network for fast adaptation.

**Constrained Optimization.** LCPO's constrained optimization formulation is structurally similar to Trust Region Policy Optimization (TRPO) (Schulman et al., 2015), despite our different problem setting.

## 4 LOCALLY-CONSTRAINED POLICY OPTIMIZATION

Our goal is to learn a policy $\pi(\cdot,\cdot)$ that takes action $a_t \sim \pi(s_t, z_t)$, in a context-driven MDP characterized by an exogenous non-stationary context process.

### 4.1 ILLUSTRATIVE EXAMPLE

Consider a simple environment with a discrete context. In this grid-world problem depicted in Figure 1, the agent can move in 4 directions in a 2D grid, and incurs a base negative reward of $-1$ per step until it reaches the terminal exit state (no penalty in the last step) or fails to reach the exit within 20 steps. The grid can be in two modes; 1) 'No Trap' mode, where the center cell is empty, and 2) 'Trap Active' mode, where walking into the center cell incurs a reward of $-10$. When in 'No Trap' mode, the optimal path passes through the center cell, and the best episodic return is $-3$. In the 'Trap Active' mode, the center cell's penalty forces the optimal path to go left at the blue cell for an optimal episodic return of $-5$. This environment mode is our discrete context and the source of non-stationarity in this simple example. The agent observes its current location and the context, i.e., whether the trap is on the grid ($z_t = 1$) or not ($z_t = 0$) in every episode (beginning from the start square).

**Advantage Actor Critic (A2C).** We use the A2C algorithm to train a policy for this environment, while its context changes every so often. Figure 1c depicts the episodic return across time and Figures 1d and 1e depict the total variation distance between the optimal and learned policy when the policy input is 'No Trap' mode (Figure 1d) or 'Trap Active' mode (Figure 1e). This distance represents how close the learned policy is to the optimal in either context. The agent initially attains optimality for the 'No Trap' mode, but once the context changes at epoch 4K it immediately forgets it. Note that during epochs 4K-16K, the A2C agent is only trained on samples from the 'Trap Active' mode, and its output for the 'No Trap' mode is drifting. When the context changes back to the 'No Trap' mode at epoch 16K, the agent behaves sub-optimally (epochs 16K-18K) before relearning. Figure 1e shows that A2C also forgets the optimal 'Trap Active' policy during the final 4K epochs.

**Key Insight.** Since the policy observes the current context $z_t$, it should be able to distinguish between different environment modes. Therefore, if the agent could surgically modify its policy on the current state-context pairs $\pi(\cdot|s_t, z_t)$ and leave outputs for other state-context pairs $\pi(\cdot|s_t, z' \neq z_t)$ unchanged, it would eventually learn a good policy for all contexts. In fact, tabular RL achieves this trivially in this

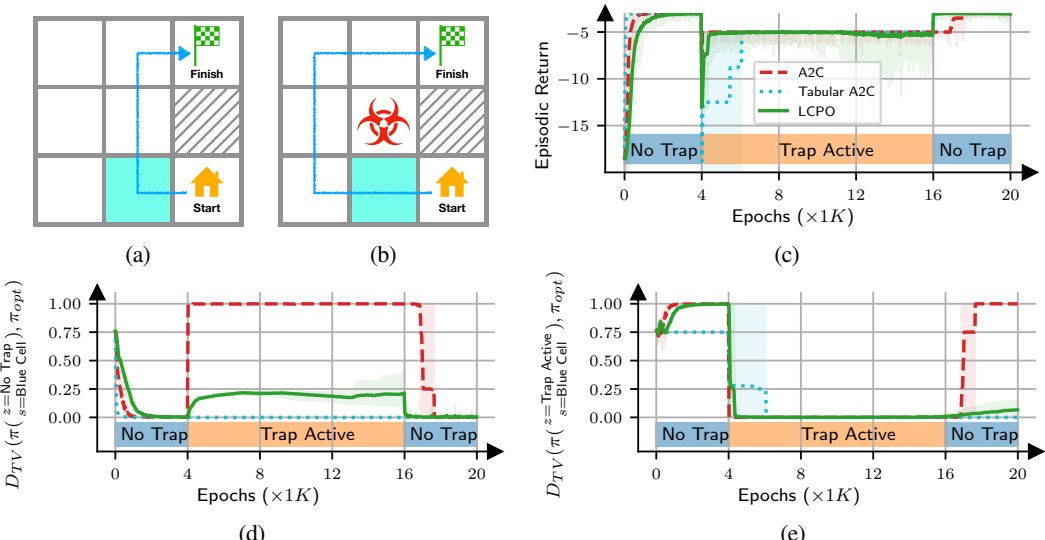

Figure 1: A 3x3 grid-world problem with two modes and the optimal path visualized in blue. **(a)** In the 'No Trap' mode, the center square is safe to pass through. **(b)** In the 'Trap' mode, the agent must avoid the trap with a longer path. **(c)** Episodic return across time in the grid environment. **(d and e)** Total variation distance between learned and optimal policy outputs for the **(d)** 'No Trap' mode, and the **(e)** 'Trap Active' mode at the blue cell (lower is better). Tabular A2C and LCPO remember the optimal decision for either context during shaded regions and instantly attain optimal returns when the environment switches.

finite discrete state-context space. To illustrate, we apply a tabular version of A2C: i.e., the policy and value networks are replaced with tables with separate rows for each state-context pair (18 total rows). Figures 1d and 1e demonstrate that the tabular RL policy for each context remains unchanged when it does not actively interact with that context. This is because when an experience is used to update the table, it only updates the row pertaining to its own state and context, and does not change rows belonging to other contexts. Under sufficient conditions, tabular RL can provably converge to the optimal policy for such environments. Due to space constraints, we state the theorem and its proof in §B in the Appendix.

Can we achieve a similar behavior with neural network function approximators? In general, updating a neural network (say, a policy network) for certain state-context pairs will change the output for all state-context pairs, leading to CF. But if we could somehow "anchor" the output of the neural network on distinct prior state-context pairs (analogous to the cells in the tabular setting) while we update the relevant state-context pairs, then the neural network would not "forget".

**LCPO.** Achieving the aforementioned anchoring does not require task labels. We only need to know if two contexts $z_i$ and $z_j$ are *different*. In particular, let the batch of recent environment interactions $<s_t, z_t, a_t, r_t>$ be $B_r$ and let all previous interactions (from possibly different contexts) be $B_a$. Suppose we have a difference detector $W(B_a, B_r)$ that can be used to sample experiences from $B_a$ that are not from the same distribution as the samples in the recent batch $B_r$, i.e., the difference detector provides out-of-distribution (OOD) samples with respect to $B_r$. Then, when optimizing the policy for the current batch $B_r$, we can constrain the policy's output on experiences sampled via $W(B_a, B_r)$ to not change (see §4.2 for details). We name this approach Locally Constrained Policy Optimization (LCPO). The result for LCPO is presented in Figures 1d and 1e. While it does not retain its policy as perfectly as tabular A2C, it does sufficiently well to recover near instantaneously upon the second switch at epoch 16K.

**Change-Point Detection (CPD) vs. OOD Detection.** CPD (and task labeling in general) requires stronger assumptions than OOD detection. The context process has to be piecewise stationary to infer task labels and context changes must happen infrequently to be detectable. Furthermore, online CPD is sensitive to outliers. In contrast, OOD is akin to defining a distance metric on the context process and can be well-defined on any context process. Consider the context process shown in Figure 2. We run this context process through a CPD algorithm (Alegre et al., 2021) for two different sensitivity factors $\sigma_{mbcd}$, and represent each detected change-point with a red vertical line. A slight increase in sensitivity leads

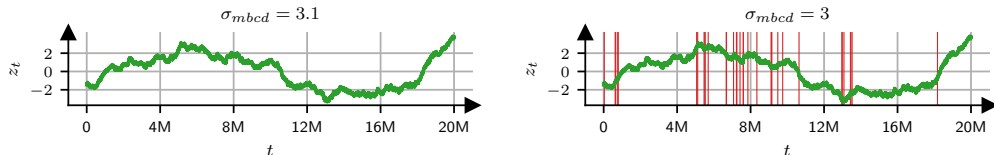

Figure 2: A sample context process $z_t$, and detected change-points at two thresholds. Teasing meaningful task boundaries is difficult for this process, but defining an OOD metric is intuitive.

to 34 detected change-points, and these change-points are also not reasonable. There is no obvious way to assign task labels for this smooth process and there aren't clearly separated segments that can be defined as tasks. However, an intuitive OOD detection method is testing if the distance between $z_i$ and $z_j$ is larger than some threshold, i.e., $|z_i - z_j| > 1$. Altogether, OOD is considerably easier in practice compared to CPD. Note that although the grid-world example – and discrete context environments in general – is a good fit for CPD, this environment was purposefully simple to explain the insight behind LCPO.

## 4.2 METHODOLOGY

Consider a parameterized policy $\pi_\theta$ with parameters $\theta$. Our task is to choose a direction for changing $\theta$ such that it improves the expected return on the most recent batch of experiences $B_r$, while the policy is 'anchored' on prior samples with sufficiently distinct context distributions, $W(B_a, B_r)$.

---

**Algorithm 1** LCPO Training

---
1:   initialize parameter vectors $\theta_0$, empty buffer $B_a$
2: **for** each iteration **do**
3:      $B_r \leftarrow$ Sample a mini-batch of new interactions
4:      $S_c \leftarrow W(B_a, B_r)$
5:      $v \leftarrow \nabla_\theta \mathcal{L}_{tot}(\theta; B_r)|_{\theta_0}$
6:      **if** $S_c$ is not empty **then**
7:          $g(x) := \nabla_\theta (x^T \nabla_\theta \mathcal{D}_{KL}(\theta_{old}, \theta; S_c)|_{\theta_0})|_{\theta_0}$
8:          $v_c \leftarrow conjgrad(v, g(\cdot))$
9:          **while** $\theta_{old} + v_c$ violates constraints **do**
10:             $v_c \leftarrow v_c/2$
11:          $\theta_0 \leftarrow \theta_0 + v_c$
12:      **else**
13:          $\theta_0 \leftarrow \theta_0 + v$
14:      $B_a \leftarrow B_a + B_r$

---

In supervised learning, this anchoring is straightforward to perform, e.g., by adding a regularization loss that directs the neural network to output the ground truth labels for OOD samples (Caruana, 1997). In the case of an RL policy, however, we do not know the ground truth (optimal actions) for anchoring the policy output. Moreover, using the actions we took in prior contexts as the ground truth is not possible, since the policy may have not converged at those times. Anchoring to those actions may cause the policy to relearn suboptimal actions from an earlier period in training. To avoid these problems, LCPO solves a constrained optimization problem that forces the policy to not change for OOD samples. Formally, we consider the following optimization problem:

$$\min_\theta \ \mathcal{L}_{tot}(\theta; B_r) \triangleq \mathcal{L}_{PG}(\theta; B_r) + \mathcal{L}_e(\theta; B_r)$$
$$s.t. \ D_{KL}(\theta_0, \theta; W(B_a, B_r)) \leq c_{anchor}$$
(1)

We use the standard definition of policy gradient loss, that optimizes a policy to maximize returns (Schulman et al., 2018; Mnih et al., 2016; Sutton & Barto, 2018):

$$\mathcal{L}_{PG}(\theta; B_r) = \mathbb{E}_{r_t \sim B_r} \left[ \sum_{t=0}^{H} -\gamma^t r_t \right]$$
(2)

We use automatic entropy regularization (Haarnoja et al., 2018c), to react to and explore in response to novel contexts. The learnable parameter $\theta_e$ is adapted such that the entropy coefficient is $e^{\theta_e}$, and the

entropy remains close to a target entropy $\bar{\mathcal{H}}$. This worked well in our experiments but LCPO could use any exploration method designed for non-stationary context-driven environments.

$$\mathcal{L}_e(\theta;B_r) = e^{\theta_e}\mathbb{E}_{s_t,z_t \sim B_r, a_t \sim \pi}[\log\pi(a_t|s_t,z_t)] \tag{3}$$

We use KL-divergence as a measure of policy change, and for simplicity we use $D_{KL}(\theta_0,\theta;W(B_a,B_r))$ as a shorthand for $\mathbb{E}_{s,z\sim W(B_a,B_r)}[D_{KL}(\pi_{\theta_0}(s,z)||\pi_\theta(s,z))]$. Here, $\theta_0$ denotes the current policy parameters, and we are solving the optimization over $\theta$ to determine the new policy parameters.

**Buffer Management.**    To avoid storing all interactions in $B_a$, we use reservoir sampling (Vitter, 1985); we randomly replace old interactions with new ones with probability $\frac{n_b}{n_s}$, where $n_b$ is the buffer size and $n_s$ is the total interactions thus far. Reservoir sampling guarantees that the interactions in the buffer are a uniformly random subset of the full set of interactions. For a pseudo-code see §C.1 in the Appendix.

**Difference detector.**    To realize $W(B_a,B_r)$, we treat it as an OOD detection task. A variety of methods can be used in practice (§5), e.g., we can compute the Mahalanobis distance (Mahalanobis, 2018) — the normalized distance of each experience's context with respect to the average context in $B_r$ — and deem any distance above a certain threshold to be OOD. To avoid a high computational overhead when sampling from $W(B_a,B_r)$, we sample a larger batch from $B_a$, and keep the state-context pairs that are OOD with respect to $B_r$. If not enough different samples exist, we do not apply the constraint for that update. For a pseudo-code and further implementation details about the OOD detector, see §C.2 and §C.3.

**Solving the constrained optimization.**    To solve this constrained optimization, we approximate the optimization goal and constraint, and calculate a search direction accordingly (pseudocode in Algorithm 1). Our problem is structurally similar to TRPO (Schulman et al., 2015), though the constraint is quite different. Similar to TRPO, we model the optimization goal with a first-order approximation, i.e., $\mathcal{L}_{tot}(\theta;\cdot) = \mathcal{L}_0 + (\theta - \theta_0)^T \nabla_\theta \mathcal{L}_{tot}(\theta;\cdot)|_{\theta_0}$, and the constraint with a second order approximation $D_{KL}(\theta_0,\theta;\cdot) = (\theta - \theta_0)^T \nabla_\theta^2 D_{KL}(\theta_0,\theta;\cdot)|_{\theta_0}(\theta - \theta_0)$. The optimization problem can therefore be written as

$$\min_\theta \ (\theta - \theta_0)^T v$$
$$s.t. \ (\theta - \theta_0)^T A(\theta - \theta_0) \le c_{anchor} \tag{4}$$

where $A_{ij} = \frac{\partial}{\partial\theta_i}\frac{\partial}{\partial\theta_j}D_{KL}(\theta_0,\theta;W(B_a,B_r))$, and $v = \nabla_\theta\mathcal{L}_{tot}(\theta;\cdot)|_{\theta_0}$. This optimization problem can be solved using the conjugate gradient method followed by a line search (Schulman et al., 2015; Achiam et al., 2017).

**Bounding policy change.**    The above formulation does not bound policy change on the current context, which could destabilize learning. We could add a second constraint, i.e., TRPO's constraint, $D_{KL}(\theta_0,\theta;B_r) \le c_{recent}$ (note that this constraint is different from that in Equation (1), as the samples come from $B_r$ instead of $W(B_a,B_r)$). However, having two second order constraints is computationally expensive. Instead, we guarantee the TRPO constraint in the line search phase (lines 9–10 in Algorithm 1), where we repeatedly decrease the gradient update size until both constraints are met.

## 5    EVALUATION

We evaluate LCPO across six environments: four from Gymnasium Mujoco (Towers et al., 2023), one from Gymnasium Classic Control (Towers et al., 2023), and a straggler mitigation task from computer systems (Mao et al., 2019a; Hamadanian et al., 2022). These environments are subject to synthetic or real context processes that affect their dynamics. Our experiments aim to answer the following questions: **(1)** How does LCPO compare to baselines, and can it perform as well as the pre-trained prescient policies (§5.1)? **(2)** How does the accuracy of the OOD sampler $W(\cdot,\cdot)$ affect LCPO (§5.2)? **(3)** How does the maximum buffer size $n_b = |B_a|$ affect LCPO (§5.3)? We include further ablations of LCPO and baselines in Appendices §E.1, §D.5 and §D.1

**Baselines.**    We consider the following approaches for comparison: **Regularization-based:** (1) Online EWC (Kirkpatrick et al., 2017; Chaudhry et al., 2018; Schwarz et al., 2018), (2) Sliding OGD (Farajtabar et al., 2019; Woo et al., 2022) and (3) Benna-Fusi DQN (BFQDN) (Kaplanis et al., 2018), **Task Inference:**

(4) Model-Based Changepoint Detection (MBCD) (Alegre et al., 2021), **Rehearsal:** (5) Model-Based Policy Optimization (MBPO) (Janner et al., 2021), (6) CLEAR (Rolnick et al., 2019), (7) PT-DQN (Anand & Precup, 2023), (8) SAC (Haarnoja et al., 2018b) and (9) Double Deep Q Network (DDQN) (Hasselt et al., 2016) **On-policy RL:** (10) A2C (Mnih et al., 2016) and (11) TRPO (single-path) (Schulman et al., 2015), both using Generalized Advantage Estimation (GAE) (Schulman et al., 2018), **Prescient RL:** (12) as described in §2, the best of policies trained with A2C (Mnih et al., 2016), TRPO (single-path) (Schulman et al., 2015), DDQN (Hasselt et al., 2016) and SAC (Haarnoja et al., 2018b). For more details about these baselines, refer to §D in the Appendix

**Experiment Setup.** We use 25 random seeds for gymnasium (5 for slower schemes) and 10 random seeds for the straggler mitigation experiments, and use the same hyperparameters for LCPO in all environments and contexts. Gym environments were modified to accept discrete action space policies, as even prescient policies struggled to learn stable continuous space policies in the presence of contexts (See §F.3). Hyperparameters and neural network structures are noted in Appendices §F.4 and §G.2. These experiments were conducted on a machine with 2 AMD EPYC 7763 CPUs (256 logical cores) and 512 GiB of RAM. With 32 concurrent runs, experiments finished in ∼1152 hours. This figure does not include runtime devoted to tuning the baselines.

**Environment and Contexts.** We consider six environments: Modified versions of Pendulum-v1 from the classic control environments, InvertedPendulum-v4, InvertedDoublePendulum-v4, Hopper-v4 and Reacher-v4 from the Mujoco environments (Towers et al., 2023), and a straggler mitigation environment (Hamadanian et al., 2022). In the gym environments, the context is an exogenous "wind" process that creates external force on joints and affects movements. We append the external wind vectors from the last 3 time-steps to the observation, since the agent cannot observe the external context that is going to be applied in the next step, and a history of prior steps helps with the prediction. We create 4 synthetic context sequences with the Ornstein–Uhlenbeck process (Uhlenbeck & Ornstein, 1930), piecewise Gaussian models, or hand-crafted signals with additive noise. These context processes cover smooth, sudden, stochastic, and predictable transitions at short horizons. All context traces are visualized in Figure 7 in the Appendix. Context traces 1 and 2 are 20 million, and context traces 3 and 4 are 8 million steps long. All baselines were allowed a 'warm-up' period of 6 million time steps, and episodes were truncated at 200 steps. For the straggler mitigation environments, we use workloads provided by the authors in (Hamadanian et al., 2022), that are from a production web framework cluster at Microsoft, collected from a single day in February 2018. These workloads are visualized in Figure 8b in the appendix.

**OOD detection.** We set the buffer size $n_b$ to 1% of all samples, which is $n_b \leq 200K$. To sample OOD state-context pairs $W(B_a, B_r)$, we use distance metrics and thresholds. For gym environments where the context is a wind process, we use L2 distance, i.e., if $\overline{w_r} = \mathbb{E}_{w \sim B_r}[w]$ is the average wind vector observed in the recent batch $B_r$, we sample a minibatch of states in $B_a$ where $W(B_a, B_r) = \{w_i | \forall w_i \in B_a : \|w_i - \overline{w_r}\|_2 > \sigma\}$. There exist domain-specific models for workload distributions in the straggler mitigation environment, but we used Mahalanobis distance as it is a well-accepted and general approach for outlier detection in prior work (Lee et al., 2018; Podolskiy et al., 2021). Concretely, we fit a Gaussian multivariate model to the recent batch $B_r$, and report a minibatch of states in $B_a$ with a Mahalanobis distance further than $\sigma$ from this distribution (see §C.2 in the Appendix for more details).

## 5.1 RESULTS

To evaluate across different gymnasium environments and traces, we score agents with *Normalized Return*, i.e., for each environment and context process we report a scaled score function where 0 and 1 are the minimum and maximum lifelong return across all agents. We prefer agents with higher scores over different environments and traces. Figure 3a provides a summary of all gymnasium experiments (full details in Tables 5 and 6 in the Appendix). LCPO maintains a lead over baselines, is close to the best-performing prescient policy, all while learning online and sequentially. We present a detailed analysis of baselines' performance in §D, and we summarize these findings below. We also report the wallclock time for each scheme in §F.2 in the appendix; LCPO is ∼1.5× as demanding as A2C.

Online EWC and Sliding OGD employ heuristics to circumvent the necessity of task labels in the original techniques. Conceptually, they implicitly assume past episodes are separate "tasks". Empirically these heuristics are not successful at solving CF. As for BFDQN, Kaplanis et al. (2018) note that while the

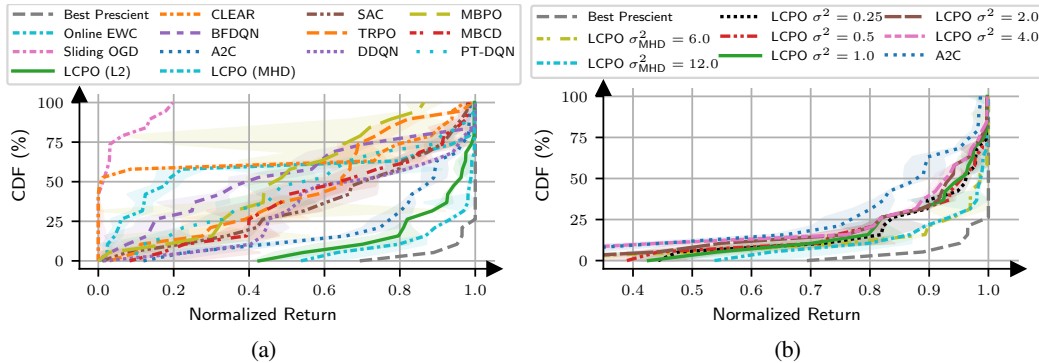

Figure 3: CDF of normalized lifelong returns, where 0/1 denote the lowest/highest returns among agents. Shaded regions denote 95% confidence intervals. **(a)** LCPO outperforms all online agents, and remains the closest to prescient policies. **(b)** LCPO is affected by the OOD threshold $\sigma$, but still outperforms baselines.

architecture was successful in simple environments, it failed with more complex and challenging ones. In our experience, this architecture did not provide any benefits compared to vanilla DDQN.

MBCD struggles to tease out meaningful task boundaries. In some experiments, it launches anywhere between 3 to 7 policies just by changing the random seed. This observation is in line with MBCD's sensitivity in §4.1, and prior work (Hamadanian et al., 2022).

MBPO performed poorly, even though we verified that its learned model of the environment is highly accurate. CLEAR (Rolnick et al., 2019) and PT-DQN (Anand & Precup, 2023) are highly hyperparameter sensitive due to how they address catastrophic forgetting. While we tuned both extensively for the Pendulum-v1 environment, as we did for all baselines, they fail catastrophically in other environments. SAC and Deep Q Network (DQN) struggle to outperform A2C in the online case. This falls in line with prior observations (Hamadanian et al., 2022) and can be attributed to the instability and hyperparameter sensitivity of off-policy RL (Duan et al., 2016; Gu et al., 2016; Haarnoja et al., 2018b) and the quick adaptation that a fully online algorithm such as A2C provides (Sutton et al., 2007). In fact, despite not having been designed for non-stationary RL, A2C is the most successful baseline.

Table 1: Tail latency (negative reward) and 95th percentile confidence ranges for different algorithms and contexts in the straggler mitigation environment.

| | | | | | | Online Learning | | | | | |
| --- | --- | --- | --- | --- | --- | --- | --- | --- | --- | --- | --- |
| | LCPO Agg | LCPO Med | LCPO Cons | MBCD | MBPO | Online EWC | A2C | TRPO | DDQN | SAC | Best Prescient |
| Workload 1 | 1070 ±10 | 1076 ±16 | 1048 ±7 | 1701 ±112 | 2531 ±197 | 2711 ±232 | 1716 ±710 | 3154 ±464 | 1701 ±47 | 1854 ±245 | 984 (TRPO) |
| Workload 2 | 589 ±43 | 617 ±62 | 586 ±27 | 678 ±38 | 891 ±54 | 724 ±22 | 604 ±109 | 864 ±105 | 633 ±7 | 644 ±27 | 509 (A2C) |

For the straggler mitigation environment, Table 1 presents the latency metric (negative reward) over two workloads. Recall that this environment uses real-world traces from a production cloud network. The overall trends are similar to the gymnasium experiments, with LCPO outperforming all other baselines. This table includes three variants of LCPO, that will be discussed further in §5.2.

## 5.2 Sensitivity to OOD metric

LCPO applies a constraint to OOD state-context pairs, as dictated by the OOD sampler $W(B_a, B_r)$. We vary the OOD threshold $\sigma$—which the OOD method uses in sampling—and monitor the normalized return for the gym environments in Figure 3b and the straggler mitigation environments in Table 1. In the gym environments, a lower value for $\sigma$ yields tighter margin of difference before a sample is deemed OOD. LCPO is affected by $\sigma$, with the smallest threshold $\sigma^2 = 0.25$ performing the best. However, LCPO still maintains a lead over the A2C baseline across $\sigma$ variations. We also experiment with a handicapped OOD metric that observes a state-context vector $x_t = \langle s_t, z_t \rangle$ *without the ability to separate state and context.*

We use the Mahalanobis distance OOD metric (Mahalanobis, 2018) at several thresholds $\sigma^2_{\text{MHD}}$ for this experiment. Despite the handicap, the LCPO +Mahalanobis surpasses the LCPO +L2 agent that we have used in this evaluation. This is not surprising, as the L2 distance is less robust than Mahalanobis distance, but easier to interpret. In the straggler mitigation environment LCPO Agg, LCPO Med and LCPO Cons use $\sigma = 5$, 6 and 7, and a higher value for $\sigma$ yields more conservative OOD samples (i.e., fewer samples are detected as OOD). The difference between these three is significant: The model in LCPO Agg allows for $26.7\times$ more samples to be considered OOD compared to LCPO Cons. Table 1 provides the normalized return for LCPO with varying thresholds, along with baselines. Notably, all variations of LCPO achieve similar results.

## 5.3 SENSITIVITY TO BUFFER SIZE

LCPO uses Reservoir sampling (Vitter, 1985) to maintain a limited number of samples $n_b$. We evaluate how sensitive LCPO is to the buffer size in Figure 4 (full results in Table 8 in the Appendix). The full experiment has 8–20M samples. LCPO maintains its high performance, even with as little as $n_b = 500$ samples, but drops below this point (statistically significant in over one third of experiments). This is not surprising, as the context traces do not change drastically at short intervals, and even 500 randomly sampled points from the trace should be enough to have a representation over all of the trace. However, with more complicated and high-dimensional contexts, a higher buffer size would likely be necessary.

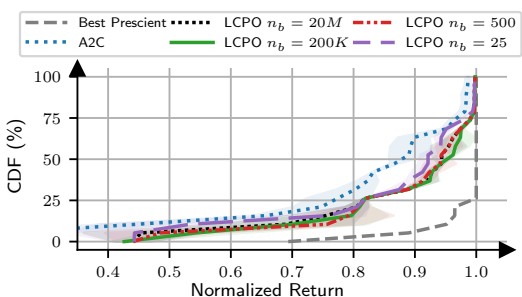

Figure 4: CDF of normalized returns of LCPO in gym environments with various buffer sizes. Shaded regions denote 95% confidence intervals. LCPO loses performance with $n_b < 500$.

## 6 DISCUSSION AND LIMITATIONS

**Network Capacity.** In general, online learning methods with bounded parameter counts will reach the function approximator's (neural network's) maximum representational capacity. LCPO is not immune from this, as we do not add parameters with more context traces. However, neither are prescient agents. To isolate the effect of this capacity and CF, we compare against prescient agents, rather than single agents trained on individual tasks or context traces (He et al., 2020). This ensures a fair evaluation that does not penalize online learning for reaching the capacity ceiling. If the maximum capacity has been reached, it may be beneficial to remove significantly old samples from $B_a$ to allow LCPO to forget such contexts, thereby favoring flexibility instead of stability.

**Exploration.** LCPO focuses on mitigating catastrophic forgetting in non-stationary RL. An orthogonal challenge in this setting is efficient exploration, i.e., to explore when the context distribution has changed but only once per new context. Our experiments used automatic entropy tuning for exploration (Haarnoja et al., 2018b); while empirically effective, this was not designed for non-stationary problems. LCPO may benefit from a better exploration methodology such as curiosity-driven exploration (Pathak et al., 2017).

**Efficient Buffer Management.** We used Reservoir Sampling (Vitter, 1985), which maintains a uniformly random buffer of all observed state-context samples so far. Future work could explore strategies that selectively store or drop samples based on their context, e.g., to maximize sample diversity.

## 7 CONCLUSION

We proposed and evaluated LCPO, a simple approach for online learning in non-stationary context-driven environments. LCPO requires two conditions: (1) the non-stationarity must be induced by an exogenous observed context process; and (2) a similarity metric is required that can inform us if two contexts come from noticeably different distributions (OOD detection). This is less restrictive than prior approaches that require either explicit or inferred task labels. Our experiments showed that LCPO outperforms baselines on several environments with real and synthetic context processes.

## REPRODUCIBILITY STATEMENT

For Theorem B.2, we include the proof and assumptions in §B. We include detailed accounts of environments, context traces, baselines, hyperparameters, software and hardware in §5 in the main text and §F and §G in the Appendix. We include implementation details and pseudo algorithms in §4.2 and §C in the Appendix. A link to the source code is also provided in §1.

## ACKNOWLEDGEMENTS

We thank our reviewers for insightful comments. This work was supported by NSF grants 1751009 and an award from the CSAIL-MSR Trustworthy AI collaboration.

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

## APPENDIX A    RELATED WORK: CONTINUED

**Meta-learning.**    Here, in the general case, the environment can switch between a set of possible MDPs without an explicit signal. At the 'meta-train' phase, the policy is allowed to train on a part or all of the MDPs. At the 'meta-test' phase, the policy must make decisions in a continual RL setup where the MDP may abruptly change, and it has to adapt to the current MDP with few-shot adaptation (Al-Shedivat et al., 2018; Nagabandi et al., 2019b;a) or context inference (Rakelly et al., 2019). In our problem setup, we assume no access to the environment beforehand.

**Multi-task RL.**    In this problem, there are several (e.g., 10 or 50) different tasks with different MDPs. The goal of this line of work is to learn a shared policy for all tasks that approaches the performance of learning separate policies per task (Yang et al., 2020; Sodhani et al., 2021). These tasks may come with contextual information about the task that can be used in the policy (Sodhani et al., 2021). This problem is not continual RL, does not experience CF, and the learner is allowed to explore all tasks at the same time. Often the goal is postitive transfer learning, i.e., learning faster on all tasks in parallel than learning tasks separately (Teh et al., 2017).

Another type of work focuses on speeding up the learning process for a set of new tasks by pre-training on a set of old tasks (Xue et al., 2024). This line of work bears similarities to multi-task RL and meta-learning.

Assuming there are no explicit signals for environment contexts, Wei & Luo (2021) provide a regret-optimal black-box RL algorithm.

**Interfernce in stationary RL.**    Investigating interference in vanilla RL (non-stationarity in sample distribution) is an adjacent and interesting line of work (Bengio et al., 2020; Pan et al., 2021; Liu et al., 2023; 2018). The type of non-stationarity discussed in these works is different than what we study. Here, non-stationarity refers to the moving target of bootstrap losses, such as Q-learning, due to shifting policies that change future data distributions. Non-stationarity in our problem setup means the MDP itself is shifting, irrespective of the changes the policy makes. The second type of non-stationarity cannot be resolved even with "perfect" RL algorithms that deal with interference in stationary MDPs.

However, there may be ideas that are transferable. For example, this line of work suggests that techniques such as GAE (Schulman et al., 2018) and target networks reduce interference (Bengio et al., 2020; Liu et al., 2023). Another interesting avenue is experimentation with sparse learning (Pan et al., 2021; Liu et al., 2018).

**Buffer limitation interference.**    A line of work, related to interference, deals with a type of non-stationarity induced by having small buffers (e.g., 32 samples, compared to the typical thousands to millions) in off-policy algorithms. These works aim to mimic an off-policy agent with unbounded buffers, and do not focus on context-driven non-stationarity. They learn policies that perform closely to unbounded agents via techniques such as following old target Q networks (Lan et al., 2023) or utilizing sparse-gradient activation functions (Lan & Mahmood, 2023). Note that we do compare to DDQN and SAC with unbounded buffers.

## APPENDIX B    PROOF OF OPTIMALITY IN TABULAR CONTEXT-DRIVEN RL

Below, we show how the non-stationary environment in §2 can be learned with vanilla RL algorithms, if the state, action and context spaces are finite, context switches occur at episode boundaries. First, we prove Lemma §B.1, and then we prove the main theorem Theorem B.2.

**Lemma B.1.** *Given a monotonically decreasing sequence* $\{\alpha_i\}_{i=1}^{\infty}$ *that satisfies the following conditions:*

$$\sum_{i=1}^{\infty} \alpha_i = \infty \qquad \sum_{i=1}^{\infty} \alpha_i^2 < \infty \qquad (5)$$

*Consider any subsequence* $\{\beta_j\}_{j=1}^{\infty}$, *where for any* $1 \leq j$, *there exists* $N \times (j-1) < i \leq N \times j$ *such that* $\beta_j = \alpha_i$.

*Prove that:*

$$\sum_{j=1}^{\infty}\beta_j=\infty \qquad \sum_{j=1}^{\infty}\beta_j^2<\infty \tag{6}$$

*Proof.* First, note that since $\{\alpha_i\}_{i=1}^{\infty}$ is monotonically decreasing, we have:

$$\alpha_{N\times i}\leq\beta_i\leq\alpha_{N\times(i-1)+1} \tag{7}$$

For the first equality, we have for all $k>0$:

$$\begin{aligned}\sum_{i=1}^{\infty}\beta_i&\geq\sum_{i=1}^{\infty}\alpha_{N\times i}\\ &\geq\sum_{i=1}^{\infty}\alpha_{N\times i+k}\end{aligned} \tag{8}$$

Thus, we have:

$$\begin{aligned}N\times\sum_{i=1}^{\infty}\beta_i&=\sum_{k=1}^{N}\sum_{i=1}^{\infty}\beta_i\\ &\geq\sum_{k=1}^{N}\sum_{i=1}^{\infty}\alpha_{N\times i+k}\\ &=\sum_{i=N+1}^{\infty}\alpha_i\end{aligned} \tag{9}$$

Therefore:

$$\sum_{i=1}^{\infty}\beta_i\geq\frac{1}{N}\sum_{i=1}^{\infty}\alpha_i-\frac{1}{N}\sum_{i=1}^{N}\alpha_i=\infty \tag{10}$$

The second bound is trivially proven:

$$\begin{aligned}\sum_{i=1}^{\infty}\beta_i^2&\leq\sum_{i=1}^{\infty}\alpha_{N\times(i-1)+1}^2\\ &\leq\sum_{i=1}^{\infty}\alpha_i^2\\ &<\infty\end{aligned} \tag{11}$$

$\square$

**Theorem B.2.** *Consider a context-driven MDP as defined in §2. Under the following set of assumptions, prove that the Q-learning algorithm (Watkins & Dayan, 1992) converges to the optimal policy:*

1. *Rewards are bounded bounded rewards $|r_t|\geq R$.*

2. *We have a a monotonically decreasing sequence of learning rates $\{\alpha_i\}_{i=1}^{\infty}$ where $0\geq\alpha_i<1$, and*

$$\sum_{i=1}^{\infty}\alpha_i=\infty \qquad \sum_{i=1}^{\infty}\alpha_i^2<\infty \tag{12}$$

3. *The state, action and context spaces are finite $|\mathcal{S}|,|\mathcal{A}|,|\mathcal{Z}|<\infty$.*

4. *There exists $N$, such that for any context $z\in\mathcal{Z}$, $z$ occurs at least once in any consecutive subsequence of the context trace of size $N$.*

5. *The context only changes at episode terminations.*

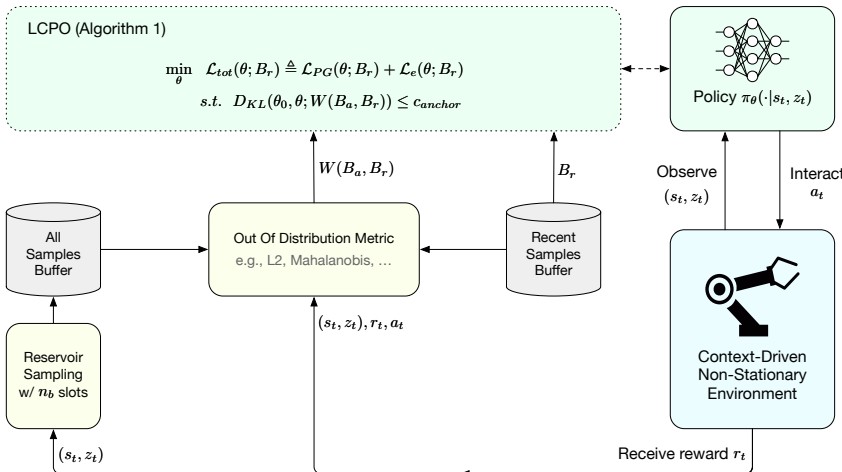

Figure 5: Architecture of LCPO.

*Proof.* As contexts only switch at episode boundaries, the Q-learning updates for each context $z \in \mathcal{Z}$ are isolated from Q-values of other contexts. Thus, for any context $z \in \mathcal{Z}$, we have a separate Q-learning session. If $I_z$ denotes the set of episode indices where the context trace is equal to $z$, If we prove

$$\sum_{i \in I_z} \alpha_i = \infty \qquad \sum_{i \in I_z} \alpha_i^2 < \infty \tag{13}$$

we can use the original Q-learning proof of convergence (Watkins & Dayan, 1992).

To prove this, note that from assumption 5 we can surmise that that for any $1 \le j$, there exists $N \times (j-1) < i \le N \times j$ such that $i \in I_z$. Therefore based on Lemma §B.1 the subsequence of learning rates satisfy the Watkins Q-learning condition. $\qquad\square$

## APPENDIX C    LCPO IMPLEMENTATION

Here, we will discuss the implementation details of LCPO. Figure 5 depicts the overall architecture of LCPO.

### C.1    RESERVOIR SAMPLING

As discussed in §4.2, LCPO needs to maintain a buffer $B_a$ of all samples observed so far. However, this buffer will grow with time and incur extensive memory costs. To limit the buffer size while maintaining a distribution of all samples observed so far, we utilize Reservoir Sampling (Vitter, 1985).

The pseudo-code for the implementation can be found in Algorithm 2. Reservoir sampling operates by maintaining a bounded list of samples $B_a$. While the number of samples in the list $B_a$ has not reached max capacity $n_b$, all samples are admitted. Once the buffer is full, a random index is sampled in the range of 0 to $n_s - 1$, where $n_s$ is the number of samples observed so far. If the index is smaller than $n_b$, the element at index $i$ is replaced with the new sample. If it is larger, the sample is thrown away. This strategy bounds the size of $B_a$, but maintains a uniform distribution of samples from the true list of all samples observed so far.

### C.2    OOD FUNCTION

LCPO requires a definition for OOD samples, i.e., samples that come from contexts far away from recent samples $B_r$. Intuitively, the optimal policy conditioned on this OOD context should be significantly different from the policy optimal conditioned on the current context.

Such metrics can be based on domain insight, where an expert who is familiar with how the context value changes the MDP would define a function to detect OOD samples. Alternatively, it may be generic OOD criteria commonly used in the literature, such as L2 thresholding, Mahalanobis distance, etc. In this case,

---

**Algorithm 2** Reservoir Sampling

---

1: **Input:** max capacity $n_b$
2: $B_a \leftarrow \{\}$ Initialize empty buffer
3: $n_s \leftarrow 0$ Initialize sample count
4: **for** each new sample $x$ **do**
5:     $n_s \leftarrow n_s + 1$ Increment the sample count
6:     **if** $n_s > n_b$ **then**
7:        $i \sim Unif(0, n_s - 1)$ Sample a random index smaller than $n_s$
8:        **if** $i < n_b$ **then**
9:           Replace element at index $i$ in $B_a$ with $x$
10:        **else**
11:           Throw new sample $x$ away
12:     **else**
13:        $B_a \leftarrow B_a + \{x\}$ Append $x$ to $B_a$

---

the OOD function needs to be tuned to be meaningful, which is a common challenge in OOD detection work. An interesting line of future work is to utilize MDP transitions in the warm-up period for tuning the threshold online.

Concretely, LCPO requires an OOD function $\omega(z, B_r)$ that denotes whether $z$ is OOD with respect to $B_r$ or not. In the L2 thresholding used for the gym environments subject to the wind context in §5, we calculate an average over $B_r$, i.e., $\mu_w = \mathbb{E}_{w \sim B_r}[w]$, and define $\omega(w, B_r) := \|w - \mu_w\|_2 > \sigma$ for some threshold $\sigma$. For the Mahalanobis distance (Mahalanobis, 2018) measure used for the straggler mitigation experiments in S5, we calculate an average and standard deviation over $B_r$, i.e., $\mu_w = \mathbb{E}_{w \sim B_r}[w]$ and $\Sigma_w = \mathbb{E}_{w \sim B_r}[(w - \mu_w)^2]$, and define $\omega(w, B_r) := (w - \mu_w)^T \Sigma_w (w - \mu_w) > \sigma^2$ for some threshold $\sigma$.

### C.3    OOD SAMPLING

Finally when want to sample a batch of size $b$ from $W(B_a, B_r)$, forming the full set $W(B_a, B_r)$ and then sampling randomly from it has a computational cost that scales with $|B_a|$. To avoid this cost, we instead sample $s$ experiences from $B_a$ and keep the ones that are OOD. If this results in at least $b$ OOD experiences, we return the samples. If not, we conclude that there aren't enough OOD samples in $B_a$ with respect to $B_r$. A pseudo-code is provided in Algorithm 3.

Analytically we can model the sampling with a binomial distribution, where $p = \mathbb{E}_{z \sim B_a}[\omega(z, B_r)]$ is the fraction of samples in $B_a$ that are OOD with respect to $B_r$. We will successfully get a batch of OOD samples with the probability $1 - F(b; s, p)$ where $F(\cdot; \cdot, \cdot)$ is the binomial cumulative distribution function. In all experiments in §5 we have set $s = 5b$. With $b = 200$, the success probability is 5% when $p = 0.18$ and 94% when $p = 0.22$. In other words, if at least 22% of the samples in $B_a$ are OOD, we are highly likely to be able to collect a batch of size $b$ of OOD samples, and unlikely if the rate is 18% or below.

---

**Algorithm 3** OOD Sampler

---

1: **Input:** All samples buffer $B_a$, Recent samples buffer $B_r$, OOD function $\omega(z, B_r)$, $s$ max sample count, $b$ batch size
2: Initialize empty list $B_{out} \leftarrow \{\}$
3: $i \leftarrow 0$
4: **while** $|B_{out}| < b$ and $i < s$ **do**
5:     $z \sim B_a$ Sample from $B_a$
6:     **if** $\omega(z, B_r)$ Sample is OOD **then**
7:        $B_{out} \leftarrow B_{out} + \{z\}$ Add sample to list
8:     $i \leftarrow i + 1$ Increment $i$
9: **if** $|B_{out}| = b$ **then**
10:     return $B_{out}$
11: **else**
12:     return $\{\}$

---

## APPENDIX D    BASELINES

### D.1    ONLINE EWC

EWC (Kirkpatrick et al., 2017) regularizes online learning with a Bayesian approach assuming task indices, and online EWC (Chaudhry et al., 2018; Schwarz et al., 2018) generalizes it to task boundaries. To adapt online EWC to our problem, we update the importance vector and average weights using a weighted moving average in every time step. The underlying learning approach is SAC (Haarnoja et al., 2018b).

EWC applies a regularization loss

$$\mathcal{L}_{ewc} = \alpha \sum_{k=1}^{N} ||\theta_t - \theta_k^*||_{F_k}^2 \tag{14}$$

to the training loss, where $N$ are the number of tasks, $\theta_k^*$ is the converged parameter set for task $k$ and $F_k$ is the diagonal of the Fisher Information Matrix (FIM) of task $k$ on the converged model for task $k$. Online EWC applies an approximation of this regularization loss

$$\mathcal{L}_{ewc} = \alpha ||\theta_t - \theta_{t-1}^*||_{F_{t-1}^*}^2 \tag{15}$$

using a running average $F_t^*$ of the diagonal of the Fisher Information Matrix (FIM), and a running average of the parameters $\theta_t^*$. The running average $F_t^*$ is updated with a weighted average $F_t^* = (1-\beta)F_{t-1}^* + \beta F_t$, where $F_t$ is the diagonal of the FIM respective to the recent parameters and samples.[1] Similarly, the running average $\theta_t^*$ uses the same parameter $\beta$.

Online EWC may constrain the policy output to remain constant on samples in the last $\approx \beta^{-1}$ epochs, but it has to strike a balance between how fast the importance metrics are updated with the newest FIM (larger $\beta$) and how long the policy has to remember its training (smaller $\beta$). This balance will ultimately depend on the context trace and how frequently they evolve. We tuned $\alpha$ and $\beta$ on Pendulum-v1 for all contexts, trying $\alpha \in \{0.05, 0.1, 0.5, 1.0, 10, 100, 1000\}$ and $\beta^{-1} \in \{1M, 3M, 10M\}$ (M denotes 1 million). The returns are visualized in Figure 6 with full details in Table 2. There is no universal $\beta$ that works well across all contexts and online EWC would not perform better than LCPO even if tuned to each context trace individually. We ultimately chose $\beta^{-1} = 3M$ samples to strike a balance across all contexts, but it struggled to even surpass SAC on other environments.

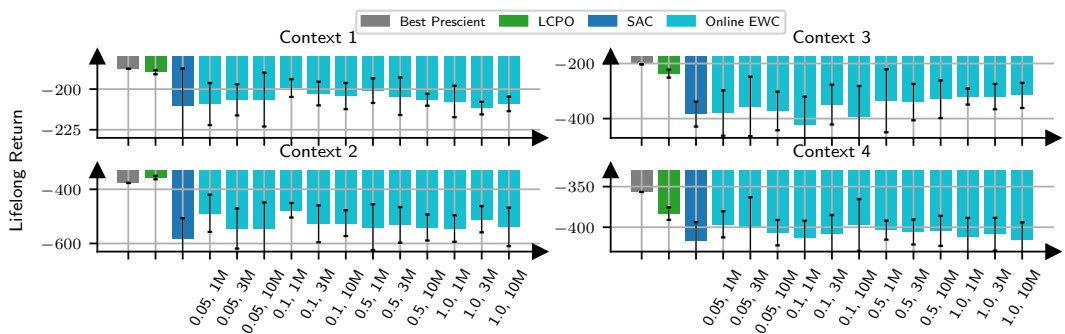

Figure 6: Pendulum-v1 lifelong returns and 95% confidence bounds of Online EWC with 12 hyperparameter trials. Hyperparameters are labeled as $\{\alpha, \beta^{-1}\}$, where $\alpha$ is the regularization strength and $\beta$ is the averaging weight. The optimal online EWC hyperparameter is sensitive to the context, but LCPO is better even if online EWC is tuned per context.

### D.2    SLIDING OGD

OGD (Farajtabar et al., 2019) projects gradients for new tasks to vector spaces that are orthogonal to previous tasks' loss functions. OGD needs task labels, and Woo et al. (2022) circumvent this by making projecting.

---

[1] We deviate from the original notations $\{\lambda, \gamma\}$ (Rusu et al., 2016), since they could be confused with the MDP discount factor $\gamma$ and GAE discount factor $\lambda$ (Schulman et al., 2018).

Table 2: Average and 95th percentile confidence ranges for lifelong returns for online EWC variants in the Pendulum-v1 environment with external wind processes.

| | **Online EWC** | | | | | | | | | | | |
| | $\alpha=0.05$ | | | $\alpha=0.1$ | | | $\alpha=0.5$ | | | $\alpha=1.0$ | | |
| $\beta^{-1}$: | $1M$ | $3M$ | $10M$ | $1M$ | $3M$ | $10M$ | $1M$ | $3M$ | $10M$ | $1M$ | $3M$ | $10M$ |
|---|---|---|---|---|---|---|---|---|---|---|---|---|
| Context Trace 1 | -209 $\pm13.0$ | -207 $\pm9.59$ | -206 $\pm16.6$ | -199 $\pm5.47$ | -203 $\pm7.28$ | -204 $\pm8.06$ | -201 $\pm7.57$ | -204 $\pm11.5$ | -207 $\pm3.67$ | -208 $\pm9.69$ | -212 $\pm3.87$ | -209 $\pm4.49$ |
| Context Trace 2 | -489 $\pm68.6$ | -545 $\pm74.1$ | -544 $\pm95.3$ | -477 $\pm26.9$ | -527 $\pm67.9$ | -525 $\pm47.6$ | -540 $\pm84.6$ | -531 $\pm65.9$ | -541 $\pm48.0$ | -545 $\pm49.0$ | -510 $\pm48.5$ | -539 $\pm71.2$ |
| Context Trace 3 | -380 $\pm82.4$ | -356 $\pm108$ | -372 $\pm70.0$ | -422 $\pm102$ | -349 $\pm72.4$ | -393 $\pm111$ | -335 $\pm114$ | -340 $\pm66.1$ | -330 $\pm68.5$ | -320 $\pm29.1$ | -320 $\pm46.1$ | -316 $\pm45.7$ |
| Context Trace 4 | -396 $\pm16.1$ | -400 $\pm36.7$ | -407 $\pm15.6$ | -413 $\pm21.1$ | -408 $\pm23.0$ | -397 $\pm31.9$ | -404 $\pm11.7$ | -406 $\pm15.4$ | -404 $\pm18.5$ | -412 $\pm23.5$ | -409 $\pm20.1$ | -416 $\pm21.7$ |

OGD (Farajtabar et al., 2019) circumvents CF by applying parameter updates that are orthogonal to the losses of past tasks. To do this, after a task has finished training, OGD calculates the gradient vector w.r.t. to samples for that task and stores those gradients. When the next task training begins, gradient updates are projected to orthogonal spaces w.r.t. the saved vectors from before. This procedure requires task labels and convergence. Sliding OGD (Woo et al., 2022) avoids needing task labels by using the gradient updates in the past $N$ episodes for projection. In other words, Sliding OGD implicitly assumes that each of the past $N$ episodes were "tasks" that have already finished training. This assumption is incorrect in our problem setup, as the context trace may change slowly. As a result, the gradient vectors of past episodes belong to the same "task" as the current episodes, and this projection fully hinders training.

### D.3 BENNA FUSI DQN

This approach applies a biologically plausible model for synapses on neural network weights in a deep Q network (Kaplanis et al., 2018). Conceptually, the weights are regularized with their past values in multiple different time scales. Kaplanis et al. (2018) note that while the Benna-Fusi DQN architecture was successful in simple environments, it failed with more complex and challenging ones. In our experience, this architecture did not provide any benefits compared to vanilla DDQN.

### D.4 MBCD

This work handles piecewise stationary environments by inferring change-points and task labels (Alegre et al., 2021). It trains models to predict environment state transitions, and launches new policies when the current model is inaccurate in predicting the state trajectory based on the CUSUM algorithm (PAGE, 1954). The underlying learning approach is SAC (Haarnoja et al., 2018b).

MBCD's sensitivity for detecting environment changes is a tunable hyperparameter; we tuned it by trying 6 values in a logarithmic space spanning $10^1$ to $10^6$ on the evaluation context traces with Pendulum-v1, and chose the best performing hyperparameter on the test set. MBCD still endlessly spawned new policies for other environments, and therefore we limited the maximum number of models to 7. Despite this, MBCD fails to perform well over the diverse set of contexts. MBCD struggles to tease out meaningful task boundaries. In some experiments, it launches anywhere between 3 to 7 policies just by changing the random seed. This observation is in line with MBCD's sensitivity in §4.1.

### D.5 MBPO

MBPO (Janner et al., 2021) is a model-based approach that trains an ensemble of experts to learn the environment model, and generates samples for an SAC (Haarnoja et al., 2018b) algorithm. If the model is accurate, it can fully replay prior contexts, thereby avoiding catastrophic forgetting.

MBPO performed poorly. If the fault is the accuracy of the learned environment models, it could be improved with approaches such online meta-learning (Finn et al., 2019) or goal-oriented model-based learning (Pong et al., 2020).. We investigated the learned models for the Pendulum-v1 experiments manually. We found these models to be very accurate, since the environment dynamics are simple.

To concretely verify that the accuracy of MBPO learned models is not the reason it underperforms, we instantiated an MBPO agent with access to the ground truth environment dynamics and context traces (but not future context traces), which we call *Ideal MBPO*. We compare the performance of Ideal MBPO vs. standard MBPO in Table 3. The performance of these two agents is widely similar. This confirms the fact that the learning algorithm itself is the problem, and not the learned models.

Table 3: Average and 95th percentile confidence ranges for lifelong returns for different algorithms and conditions in the Pendulum-v1 environment with external wind processes. An MBPO agent with access to the ground truth model performs similarly to the MBPO model. Schemes with superiority beyond 95% confidence are highlighted in bold.

| | Online Learning | | | | |
| --- | --- | --- | --- | --- | --- |
| | LCPO | MBPO | Ideal MBPO | A2C | Best Prescient |
| Context Trace 1 | **-190** $\pm$**0.45** | -325 $\pm$47.2 | -320 $\pm$44.2 | -201 $\pm$3.92 | -187 (SAC) |
| Context Trace 2 | **-355** $\pm$**2.57** | -843 $\pm$24.2 | -870 $\pm$56.4 | -377 $\pm$8.05 | -376 (DDQN) |
| Context Trace 3 | **-240** $\pm$**4.41** | -603 $\pm$213 | -637 $\pm$90.0 | -307 $\pm$32.6 | -203 (SAC) |
| Context Trace 4 | **-378** $\pm$**3.02** | -553 $\pm$90.9 | -555 $\pm$84.1 | -399 $\pm$7.76 | -357 (SAC) |

The reason is the way that MBPO samples experiences for training. At every iteration, MBPO samples a batch of actual interactions from its experience buffer and generates hypothetical interactions from them. These hypothetical interactions amass in a second buffer, which is used to train an SAC agent. During the course of training, the second buffer accumulates more interactions generated from samples from the start of the experiment compared to recent samples. This is not an issue when the problem is stationary, but in non-stationary RL subject to an context process, this leads to over-emphasis of the context processes encountered earlier in the experiment. As such, MBPO fails to even surpass SAC. Prior work has observed the sensitivity of off-policy approaches to such sampling strategies (Isele & Cosgun, 2018; Hamadanian et al., 2022).

## D.6 CLEAR

This approach aims to mitigate CF with off-policy learning and maintain quick adaptation with on-policy learning (Rolnick et al., 2019). They use IMPALA and V-trace (Espeholt et al., 2018) on recent batches for on-policy and stale batches for off-policy RL.

CLEAR (Rolnick et al., 2019) aims to mix the quick adaptation of on-policy RL and the CF resilience of off-policy RL, but in practice this fusion makes it very hyperparameter sensitive. The V-trace algorithm was originally intended to correct for lagging policies in a distributed RL architecture (Espeholt et al., 2018). V-trace uses clipped importance sampling to correct for the drift between the logging and training policies, which reduces variance but biases the loss. With small lags between workers, the bias is small. If V-trace is used in a scenario where the logging and training policy are very different, such as in CLEAR (Rolnick et al., 2019), the bias becomes significant enough to hinder training. CLEAR attempts to circumvent this by inducing a regularization loss on the actor and critic. Yet, this regularization will count against improving the RL policy for better returns, and will be brittle. The correct balance between policy improvement and this regularization will ultimately depend on the environment and context trace. While we tuned CLEAR extensively for the Pendulum-v1 environment, as we did for all baselines, it fails catastrophically in other environments.

## D.7 PT-DQN

PT-DQN (Anand & Precup, 2023) learns two separate networks with two different goals; (1) a permanent network that is updated infrequently and slowly, and aims to learn a generalized estimate of Q-values in various tasks, and (2) a transient network that learns *and forgets* aggressively, and aims to quickly learn the optimal policy for the current task. Anand & Precup (2023) prove PT-DQN asymptotically converges to the optimum predictors in piecewise stationary prediction tasks with tabular input spaces.

Note that PT-DQN avoids catastrophic forgetting indirectly by slowly updating a permanent Q-network. The hope is that it strikes the right balance in learning this network slowly enough such that earlier contexts are not forgotten, but updates it frequently enough such that new knowledge is not lost. This trade-off is brittle; how fast the transient Q-network should forget and relearn, and how slowly the permanent Q-network should be updated highly depends on (1) how quickly the context process changes, and (2) by how much these changes affect the transition dynamics of the MDP. As was also observed with CLEAR (§D.6), balances of this nature are brittle, due to the indirect nature of how these techniques address catastrophic forgetting.

Thus, PT-DQN is understandly hyperparameter sensitive. Indeed, to tune PT-DQN on Pendulum-v1 environment, (as done for all baselines), we carried out three rounds of grid-search on five hyperparameters, totalling 440 different combinations. Despite PT-DQN being competitive with DDQN on Pendulum-v1, the performance is unpredictable in other environments.

### D.8 OFF-POLICY RL

Off-policy RL is potentially capable of overcoming CF due to replay buffers, at the cost of unstable training. We consider DDQN (Hasselt et al., 2016) and SAC (with automatic entropy regularization, similar to LCPO) (Haarnoja et al., 2018b).

### D.9 ON-POLICY RL

On-policy RL is susceptible to CF, but more stable in online learning compared to off-policy RL algorithms, and the fast adaptation of these algorithms can also help them 'track' the optimal policy as the environment changes (Sutton et al., 2007). We compare with A2C (Mnih et al., 2016) and TRPO (single-path) (Schulman et al., 2015), with GAE (Schulman et al., 2018) applied. Note that TRPO vine is not possible in online learning, as it requires rolling back the environment world.

### D.10 PRESCIENT RL

To establish an upper-bound on the best performance that an online learner can achieve, we train *prescient policies*, as discussed in §2. We allow prescient policies to have unlimited access to the contexts and environment dynamics, i.e., they are able to replay any particular environment and context as many times as necessary. Since prescient policies can interact with multiple contexts in parallel during training, they do not suffer from CF. In contrast, all other baselines (and LCPO) are only allowed to experience contexts sequentially as they occur over time and must adapt the policy on the go. We report results for the best of four prescient policies with the following model-free algorithms: A2C (Mnih et al., 2016), TRPO (single-path) (Schulman et al., 2015), DDQN (Hasselt et al., 2016) and SAC (Haarnoja et al., 2018b).

## APPENDIX E  LCPO VARIANTS

In §4.2 we discussed our main approach to solving the constrained optimization problem below:

$$
\begin{aligned}
\min_{\theta} \ & \mathcal{L}_{PG}(\theta;B_r) \\
s.t. \ & D_{KL}(\theta_0,\theta;W(B_a,B_r)) \leq c_{anchor}
\end{aligned}
\tag{16}
$$

The goal is to optimize the policy gradient loss, i.e., maximize returns on current input, while minimizing policy change on past observed inputs with the *anchoring constraint*. We outlined a solution to this problem, with a pseudo-code in Algorithm 1, that is essentially a second order constrained optimization plus a line search phase:

$$
\begin{aligned}
\min_{\theta} \ & (\theta-\theta_0)^T v_{PG} \\
s.t. \ & (\theta-\theta_0)^T A(\theta-\theta_0) \leq c_{anchor}
\end{aligned}
\tag{17}
$$

where $A_{ij} = \frac{\partial}{\partial \theta_i}\frac{\partial}{\partial \theta_j}D_{KL}(\theta_0,\theta;W(B_a,B_r))$, and $v_{PG} = \nabla_\theta \mathcal{L}_{PG}(\theta;\cdot)|_{\theta_0}$. However, there is another way to solve this problem, that we discuss as follows.

### E.1    LCPO-P

An alternative way to uphold the anchoring constraint is to directly add it as term in the loss function. Let us define:

$$\mathcal{L}_{anchor}(\theta,\theta_0;B_r,B_a)=$$
$$\mathbb{E}_{s,z\sim W(B_a,B_r)}[CELoss(\pi_{\theta_0}(s,z),\pi_\theta(s,z))] \tag{18}$$

Where we use the Cross Entropy loss to incentivize policy anchoring. Then, we optimize the following total loss:

$$\min_\theta \ \mathcal{L}_{PG}(\theta;\cdot)+\kappa.\mathcal{L}_{anchor}(\theta,\theta_0;B_r,B_a) \tag{19}$$

This approach is even less compute intensive than LCPO, but is not possible in vanilla policy gradient. This is because the gradient direction from $\mathcal{L}_{anchor}$ is zero when $\theta=\theta_0$ and will not affect the optimization. Therefore, we have to repeat the gradient update several times before this term has an effect. The optimization setup in Proximal Policy Optimization (PPO) (Schulman et al., 2017) allows for several gradient steps with one batch of data, and therefore we apply the above loss in the PPO optimization step. We dub this approach LCPO-P (P stands for proximal).

Table 4: Average and 95th percentile confidence ranges for lifelong returns for LCPO and LCPO-P and conditions in the Pendulum-v1 environment with external wind processes. Schemes with superiority beyond 95% confidence are highlighted in bold.

|  | LCPO | LCPO-P | Best Baseline | **Best Prescient** |
|---|---|---|---|---|
| Context Trace 1 | **-190** $\pm$**0.45** | -211 $\pm$3.92 | -201 (A2C) $\pm$3.92 | -187 (SAC) |
| Context Trace 2 | **-355** $\pm$**2.57** | -388 $\pm$7.33 | -377 (A2C) $\pm$8.05 | -376 (DDQN) |
| Context Trace 3 | **-240** $\pm$**4.41** | -271 $\pm$21.9 | -262 (CLEAR) $\pm$2.2 | -203 (SAC) |
| Context Trace 4 | **-378** $\pm$**3.02** | -395 $\pm$9.32 | -399 (A2C) $\pm$7.76 | -357 (SAC) |

We compare LCPO and LCPO-P in Table 4. We tuned $\kappa$ on the Pendulum-v1 environment, similar to all baselines, and finalized on $\kappa=10$. Despite this, LCPO-P fail to outperform the best baseline for each context trace even on Pendulum-v1. The stark contrast between LCPO and LCPO-P is due to the difficulty in tuning $\kappa$. LCPO's optimization constraint guarantees that the policy is anchored on past contexts, while the loss term added in LCPO-P motivates for this change to be small. Although the setup in LCPO-P is the Lagrangian dual of the setup in LCPO, they are only equivalent when $\kappa$ is tuned properly per each gradient step.

## APPENDIX F    LOCOMOTION TASKS IN GYMNASIUM

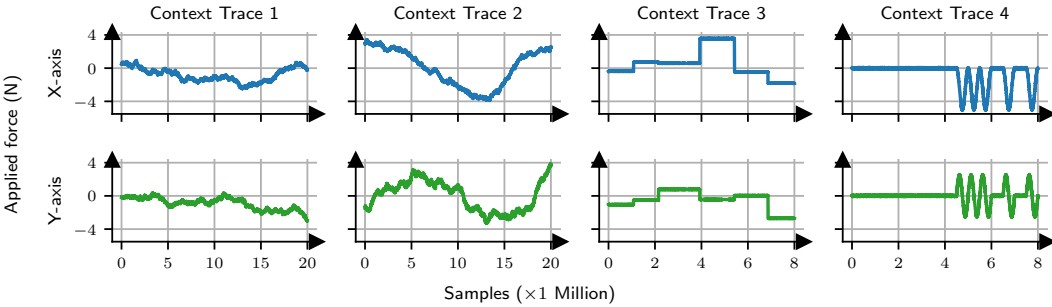

Figure 7: External wind force, per axis and context trace.

## F.1 FULL RESULTS

Tables 5 and 6 presents the lifelong return for all agents, environments and context traces. Table 7 presents the lifelong return for multiple LCPO agents with different OOD thresholds $\sigma$. Table 8 presents the lifelong return for multiple LCPO agents with different buffer sizes $n_b$.

Table 5: Average and 95th percentile confidence ranges for lifelong return for different algorithms and conditions in environments with external wind processes. Schemes with superiority beyond 95% confidence are highlighted in bold. **Continued in Table 6.**

| | | Online Learning | | | | | | Prescient Policies | | | |
| | | LCPO | Online EWC | Sliding OGD | CLEAR | MBCD | MBPO | A2C | TRPO | DDQN | SAC |
|---|---|---|---|---|---|---|---|---|---|---|---|
| Pendulum | Trace 1 | -190 ±0.45 | -204 ±2.37 | -1195 ±4.38 | -215 ±0.42 | -207 ±23.4 | -325 ±47.2 | -208 | -197 | -188 | **-187** |
| | Trace 2 | **-355** ±**2.57** | -525 ±22.6 | -1232 ±3.25 | -429 ±1.58 | -577 ±71.3 | -843 ±24.2 | -407 | -399 | **-376** | -380 |
| | Trace 3 | **-240** ±**4.41** | -345 ±29.6 | -1181 ±3.78 | -262 ±2.20 | -349 ±28.9 | -603 ±213 | -224 | -320 | -228 | **-203** |
| | Trace 4 | **-378** ±**3.02** | -412 ±6.52 | -1233 ±4.41 | -411 ±2.45 | -430 ±15.4 | -553 ±90.9 | -383 | -381 | -384 | **-357** |
| Inverse Pendulum | Trace 1 | 18.7 ±2.85 | 3.47 ±0.32 | 2.21 ±0.01 | 2.28 ±0.10 | 6.17 ±6.10 | 2.88 ±1.78 | 24.4 | **25.7** | 24.1 | 24.8 |
| | Trace 2 | **5.31** ±**0.54** | 1.55 ±0.03 | 1.58 ±0.00 | 1.60 ±0.03 | 2.10 ±1.06 | 1.52 ±0.01 | 7.86 | **8.51** | 7.18 | 7.75 |
| | Trace 3 | 15.4 ±0.22 | 5.56 ±0.88 | 4.24 ±0.02 | 2.45 ±0.36 | 14.2 ±0.77 | 14.4 ±0.23 | 15.5 | **16.5** | 15.1 | 14.3 |
| | Trace 4 | 92.4 ±0.19 | 5.43 ±0.88 | 4.64 ±0.01 | 2.87 ±0.17 | 46.5 ±4.85 | 37.6 ±0.98 | 102 | **112** | 53.7 | 40.1 |
| Inverse Double Pendulum | Trace 1 | 114 ±3.54 | 38.8 ±2.15 | 38.5 ±0.10 | 34.2 ±0.46 | 77.3 ±13.5 | 68.5 ±19.3 | 142 | **165** | 88.0 | 95.5 |
| | Trace 2 | 56.2 ±2.28 | 30.1 ±1.03 | 30.9 ±0.03 | 28.9 ±0.20 | 44.2 ±3.27 | 37.2 ±9.92 | 51.4 | **64.0** | 46.8 | 49.5 |
| | Trace 3 | 65.1 ±0.98 | 36.1 ±1.80 | 36.4 ±0.06 | 27.1 ±0.66 | 45.8 ±2.38 | 42.7 ±1.55 | 63.5 | **74.0** | 63.6 | 47.4 |
| | Trace 4 | 89.2 ±0.33 | 37.7 ±3.22 | 41.4 ±0.07 | 29.3 ±0.80 | 75.0 ±4.65 | 59.5 ±2.86 | 91.9 | **96.9** | 90.7 | 78.0 |
| Hopper | Trace 1 | **244** ±**5.58** | 34.1 ±5.05 | 11.4 ±0.42 | 3.92 ±0.54 | 153 ±33.2 | 130 ±12.8 | **240** | 215 | 155 | 184 |
| | Trace 2 | **144** ±**3.44** | 28.2 ±4.98 | 12.0 ±0.53 | 2.87 ±0.25 | 76.0 ±15.3 | 66.0 ±18.2 | 128 | **131** | 86.6 | 96.7 |
| | Trace 3 | **508** ±**2.54** | 27.0 ±10.4 | 19.3 ±0.60 | 3.62 ±0.31 | 240 ±81.7 | 264 ±49.9 | **520** | 492 | 327 | 308 |
| | Trace 4 | **296** ±**2.59** | 22.2 ±8.47 | 12.8 ±0.38 | 2.94 ±0.10 | 127 ±27.3 | 205 ±37.5 | **319** | 297 | 189 | 189 |
| Reacher | Trace 1 | -26.5 ±0.21 | -16.3 ±0.60 | -6622 ±124 | -1780 ±958 | -169 ±122 | -1862 ±2255 | -8.00 | -7.59 | **-7.59** | -14.3 |
| | Trace 2 | -19.8 ±0.11 | -16.4 ±0.37 | -6733 ±74.0 | -1695 ±731 | -339 ±493 | -4308 ±3764 | -8.33 | -3866 | **-7.41** | -15.3 |
| | Trace 3 | -27.3 ±0.89 | -15.4 ±0.23 | -6722 ±131 | -799 ±161 | -596 ±857 | -2058 ±4173 | -9.63 | -8.42 | **-8.41** | -15.7 |
| | Trace 4 | -29.4 ±0.29 | -16.6 ±0.44 | -6718 ±102 | -1392 ±1060 | -121 ±161 | -2354 ±2988 | -9.63 | -8.57 | **-7.97** | -15.7 |

## F.2 COMPUTATIONAL LOAD

LCPO is about $1.5\times$ more computationally demanding than the leading baseline A2C. Table 9 depicts the total runtime per each environment interaction in the experiments in §5.1.

Table 6: Average and 95th percentile confidence ranges for lifelong return for different algorithms and conditions in environments with external wind processes. Schemes with superiority beyond 95% confidence are highlighted in bold. **Continued from Table 5.**

| | | Online Learning | | | | | | | Prescient Policies | | | |
|---|---|---|---|---|---|---|---|---|---|---|---|---|
| | | LCPO | A2C | TRPO | BFDQN | PT-DQN | DDQN | SAC | A2C | TRPO | DDQN | SAC |
| Pendulum | Trace 1 | -190 ±0.45 | -201 ±3.92 | -222 ±4.14 | -346 ±15.0 | -205 ±2.80 | -219 ±8.88 | -207 ±4.12 | -208 | -197 | -188 | **-187** |
| | Trace 2 | **-355** ±2.57 | -377 ±8.05 | -548 ±41.3 | -709 ±64.9 | -835 ±120 | -756 ±103 | -636 ±62.3 | -407 | -399 | **-376** | -380 |
| | Trace 3 | **-240** ±4.41 | -307 ±32.6 | -511 ±63.4 | -766 ±75.1 | -596 ±79.5 | -612 ±76.2 | -402 ±31.0 | -224 | -320 | -228 | **-203** |
| | Trace 4 | **-378** ±3.02 | -399 ±7.76 | -648 ±31.4 | -693 ±53.3 | -436 ±32.0 | -409 ±7.59 | -418 ±5.52 | -383 | -381 | -384 | **-357** |
| Inverse Pendulum | Trace 1 | 18.7 ±2.85 | 7.57 ±2.31 | 4.61 ±1.84 | 5.79 ±1.13 | 13.6 ±2.27 | 12.7 ±1.38 | 3.97 ±0.67 | 24.4 | **25.7** | 24.1 | 24.8 |
| | Trace 2 | **5.31** ±0.54 | 2.36 ±0.45 | 1.73 ±0.28 | 1.74 ±0.09 | 2.24 ±0.13 | 1.60 ±0.06 | 1.59 ±0.06 | 7.86 | **8.51** | 7.18 | 7.75 |
| | Trace 3 | 15.4 ±0.22 | 15.0 ±0.46 | 7.94 ±2.36 | 4.28 ±1.58 | 15.0 ±0.17 | 14.9 ±0.14 | 14.7 ±0.05 | 15.5 | **16.5** | 15.1 | 14.3 |
| | Trace 4 | 92.4 ±0.19 | **93.7** ±0.21 | 75.9 ±10.6 | 79.6 ±2.88 | 75.5 ±3.18 | 92.1 ±1.18 | 50.5 ±1.46 | 102 | **112** | 53.7 | 40.1 |
| Inverse Double Pendulum | Trace 1 | 114 ±3.54 | 118 ±9.47 | 86.3 ±13.9 | 48.7 ±7.99 | 76.6 ±1.91 | 80.0 ±0.95 | 75.6 ±5.67 | 142 | **165** | 88.0 | 95.5 |
| | Trace 2 | 56.2 ±2.28 | 54.3 ±1.72 | 30.0 ±3.60 | 25.6 ±0.08 | 43.6 ±1.52 | 42.4 ±1.17 | 38.8 ±2.37 | 51.4 | **64.0** | 46.8 | 49.5 |
| | Trace 3 | 65.1 ±0.98 | 63.4 ±2.37 | 49.7 ±3.93 | 39.0 ±4.49 | 61.9 ±0.29 | 61.6 ±0.26 | 46.8 ±0.23 | 63.5 | **74.0** | 63.6 | 47.4 |
| | Trace 4 | 89.2 ±0.33 | 87.9 ±0.53 | 75.9 ±1.85 | 67.6 ±4.37 | 93.9 ±0.16 | 93.9 ±0.18 | 78.7 ±0.63 | 91.9 | **96.9** | 90.7 | 78.0 |
| Hopper | Trace 1 | **244** ±5.58 | 203 ±7.73 | 164 ±7.25 | 93.5 ±3.24 | 86.0 ±13.9 | 136 ±8.83 | 154 ±14.1 | **240** | 215 | 155 | 184 |
| | Trace 2 | **144** ±3.44 | 97.9 ±3.07 | 65.8 ±9.11 | 23.2 ±6.85 | 30.0 ±7.43 | 68.1 ±4.11 | 81.7 ±5.69 | 128 | **131** | 86.6 | 96.7 |
| | Trace 3 | **508** ±2.54 | 484 ±13.8 | 443 ±17.1 | 137 ±12.2 | 52.2 ±6.63 | 220 ±6.15 | 360 ±39.2 | **520** | 492 | 327 | 308 |
| | Trace 4 | **296** ±2.59 | 263 ±5.59 | 186 ±14.9 | 105 ±6.14 | 117 ±13.7 | 161 ±8.02 | 258 ±14.4 | **319** | 297 | 189 | 189 |
| Reacher | Trace 1 | -26.5 ±0.21 | -118 ±45.9 | -2346 ±471 | -9.41 ±0.15 | -2492 ±1898 | -7.84 ±0.03 | -370 ±169 | -8.00 | -7.59 | **-7.59** | -14.3 |
| | Trace 2 | -19.8 ±0.11 | -102 ±34.6 | -4728 ±515 | -11.5 ±3.52 | -19.9 ±17.5 | -7.78 ±0.07 | -618 ±290 | -8.33 | -3866 | **-7.41** | -15.3 |
| | Trace 3 | -27.3 ±0.89 | -133 ±61.5 | -71.7 ±80.0 | -10.1 ±0.32 | -5916 ±2335 | -8.56 ±0.04 | -258 ±140 | -9.63 | -8.42 | **-8.41** | -15.7 |
| | Trace 4 | -29.4 ±0.29 | -117 ±56.3 | -1780 ±272 | -9.17 ±0.45 | -23.2 ±10.4 | -9.21 ±1.22 | -157 ±59.6 | -9.63 | -8.57 | **-7.97** | -15.7 |

## F.3 ACTION SPACE

Pendulum-v1 and Mujoco environments by default have continuous action spaces. We observed instability while learning policies with continuous policy classes even for the prescient policies, and were concerned about how this can affect the validity of our online experiments, which are considerably more challenging. As the action space is tangent to our problem, we discretized each dimension of the action space to 15 atoms, spaced equally from the minimum to the maximum action in each dimension. This stabilized training greatly, and is not surprising, as past work (Tang & Agrawal, 2019) supports this observation. The reward metric, continuous state space and truncation and termination conditions remain unchanged.

We provide the achieved episodic return for all baselines in §5 in Table 10, over 10 seeds for the Pendulum-v1 envioronment, which we can compare to SB3 (Raffin et al., 2021) and RL-Zoo (Raffin, 2020) reported figures. These experiments finished in approximately 46 minutes. A2C, DDQN and SAC were trained for

Table 7: Average and 95th percentile confidence ranges for lifelong return in LCPO with different OOD metrics and thresholds and other agents for different conditions in environments with external wind processes. L2 stands for the L2-distance OOD metric and MHD stands for the Mahalanobis distance OOD metric. Schemes with superiority beyond 95% confidence are highlighted in bold (LCPO agents are only compared against baselines).

| | | LCPO (L2) | | | | | LCPO (L2) | | | |
|---|---|---|---|---|---|---|---|---|---|---|
| | | $\sigma^2=0.25$ | $\sigma^2=0.5$ | $\sigma^2=1$ | $\sigma^2=2$ | $\sigma^2=4$ | $\sigma^2_{\text{MHD}}=6$ | $\sigma^2_{\text{MHD}}=12$ | Best Baseline | Best Prescient |
| Pendulum | Trace 1 | -189 ±0.52 | -190 ±0.66 | -190 ±0.45 | -194 ±0.62 | -199 ±0.32 | -188 ±0.50 | **-188** ±**0.53** | -201 (A2C) ±3.92 | -187 (SAC) |
| | Trace 2 | -349 ±2.12 | -354 ±2.68 | -355 ±2.57 | -359 ±2.64 | -361 ±3.03 | **-346** ±**1.66** | -346 ±2.49 | -377 (A2C) ±8.05 | -376 (DDQN) |
| | Trace 3 | -242 ±5.34 | -242 ±5.34 | -240 ±4.41 | -235 ±3.94 | **-231** ±**4.13** | -233 ±4.39 | -233 ±4.40 | -262 (CLEAR) ±2.20 | -203 (SAC) |
| | Trace 4 | -375 ±2.76 | -377 ±4.12 | -378 ±3.02 | -380 ±2.99 | -385 ±2.78 | **-369** ±**3.45** | -373 ±3.07 | -399 (A2C) ±7.76 | -357 (SAC) |
| Inverse Pendulum | Trace 1 | **21.5** ±**1.72** | 18.9 ±3.03 | 18.7 ±2.85 | 6.05 ±2.13 | 2.77 ±0.16 | 4.54 ±2.64 | 17.1 ±2.96 | 13.6 (PT-DQN) ±2.27 | 25.7 (TRPO) |
| | Trace 2 | 4.85 ±0.60 | 4.24 ±0.71 | 5.31 ±0.54 | **5.35** ±**0.56** | 2.79 ±0.71 | 5.12 ±0.68 | 5.28 ±0.62 | 2.36 (A2C) ±0.45 | 8.51 (TRPO) |
| | Trace 3 | 15.1 ±0.36 | 15.6 ±0.13 | 15.4 ±0.22 | 15.4 ±0.23 | 15.4 ±0.23 | **16.3** ±**0.03** | 16.3 ±0.04 | 15.0 (A2C) ±0.46 | 16.5 (TRPO) |
| | Trace 4 | 94.5 ±0.24 | 92.4 ±0.19 | 92.4 ±0.19 | 92.4 ±0.19 | 92.4 ±0.19 | **100** ±**0.25** | 97.6 ±0.39 | 93.7 (A2C) ±0.21 | 112 (TRPO) |
| Inverse Double Pendulum | Trace 1 | 118 ±5.26 | 118 ±5.26 | 114 ±3.54 | 118 ±5.26 | 118 ±5.26 | **223** ±**5.39** | 221 ±6.53 | 118 (A2C) ±9.47 | 165 (TRPO) |
| | Trace 2 | 53.4 ±3.35 | 54.2 ±1.60 | 56.2 ±2.28 | 54.9 ±2.23 | 54.9 ±2.23 | 62.9 ±2.95 | **63.3** ±**4.44** | 54.3 (A2C) ±1.72 | 64.0 (TRPO) |
| | Trace 3 | 65.5 ±0.92 | 65.0 ±0.91 | 65.1 ±0.98 | 64.9 ±0.91 | 64.9 ±0.91 | 62.5 ±4.07 | 64.5 ±3.41 | 63.4 (A2C) ±2.37 | 74.0 (TRPO) |
| | Trace 4 | 89.9 ±0.27 | 89.4 ±0.24 | 89.2 ±0.33 | 89.4 ±0.24 | 89.4 ±0.24 | 90.1 ±0.13 | 89.7 ±0.14 | **93.9** (DDQN) ±**0.18** | 96.9 (TRPO) |
| Hopper | Trace 1 | 247 ±6.96 | 250 ±8.38 | 244 ±5.58 | 235 ±2.81 | 222 ±3.49 | 245 ±10.2 | **252** ±**8.11** | 203 (A2C) ±7.73 | 240 (A2C) |
| | Trace 2 | 144 ±3.48 | 144 ±3.20 | 144 ±3.44 | 139 ±3.14 | 138 ±3.50 | **147** ±**3.59** | 146 ±3.79 | 97.9 (A2C) ±3.07 | 131 (TRPO) |
| | Trace 3 | 520 ±4.01 | 512 ±2.46 | 508 ±2.54 | 508 ±2.46 | 508 ±2.46 | **539** ±**8.78** | 533 ±6.40 | 484 (A2C) ±13.8 | 520 (A2C) |
| | Trace 4 | **302** ±**3.26** | 298 ±3.22 | 296 ±2.59 | 292 ±2.04 | 291 ±4.09 | 300 ±3.98 | 298 ±2.88 | 263 (A2C) ±5.59 | 319 (A2C) |
| Reacher | Trace 1 | -26.7 ±0.18 | -26.5 ±0.19 | -26.5 ±0.21 | -24.5 ±0.32 | -21.6 ±0.19 | -25.1 ±0.12 | -25.1 ±0.09 | **-7.84** (DDQN) ±**0.03** | -7.59 (DDQN) |
| | Trace 2 | -25.5 ±0.11 | -22.6 ±0.13 | -19.8 ±0.11 | -19.8 ±0.12 | -19.8 ±0.12 | -23.9 ±0.23 | -23.8 ±0.22 | **-7.78** (DDQN) ±**0.07** | -7.41 (DDQN) |
| | Trace 3 | -27.0 ±1.07 | -27.0 ±1.07 | -27.3 ±0.89 | -29.0 ±0.62 | -28.4 ±0.67 | -25.4 ±0.82 | -25.5 ±0.56 | **-8.56** (DDQN) ±**0.04** | -8.41 (DDQN) |
| | Trace 4 | -30.6 ±0.29 | -30.4 ±0.36 | -29.4 ±0.29 | -27.7 ±0.37 | -26.0 ±0.34 | -24.6 ±0.19 | -24.2 ±0.23 | **-9.17** (BFDQN) ±**0.45** | -7.97 (DDQN) |

8000 epochs, and TRPO was trained for 500 epochs. Evaluations are on 1000 episodes. As these results show, the agents exhibit stable training with a discretized action space.

## F.4 EXPERIMENT SETUP

We use Gymnasium (v0.29.1, MIT license) and Mujoco (v3.1.1, Apache-2.0 license). Our baseline and LCPO implementations use the Pytorch (Paszke et al., 2019) (v1.13.1, BSD-style license) library. Table 11 is a comprehensive list of all hyperparameters used in training and the environment.

All baselines were tuned on Pendulum-v1 via a multi-phased grid search, similar to that in §D.1. General parameters such as discount horizon were copied from the base RL algorithm each baseline is using (e.g.,

Table 8: Average and 95th percentile confidence ranges for lifelong return in LCPO with multiple buffer sizes and other agents for different conditions in environments with external wind processes. Schemes with superiority beyond 95% confidence are highlighted in bold (LCPO agents are only compared against baselines).

| | | LCPO | | | | Best Baseline | Best Prescient |
|---|---|---|---|---|---|---|---|
| | | $n_b=20M$ | $n_b=200K$ | $n_b=500$ | $n_b=25$ | | |
| Pendulum | Trace 1 | -191 ±0.65 | **-190** **±0.45** | -190 ±0.62 | -193 ±0.99 | -201 (A2C) ±3.92 | -187 (SAC) |
| | Trace 2 | -355 ±2.02 | **-355** **±2.57** | -356 ±2.01 | -366 ±3.57 | -377 (A2C) ±8.05 | -376 (DDQN) |
| | Trace 3 | **-240** **±6.83** | -240 ±4.41 | -240 ±9.10 | -259 ±10.7 | -262 (CLEAR) ±2.20 | -203 (SAC) |
| | Trace 4 | -379 ±3.28 | **-378** **±3.02** | -379 ±3.78 | -400 ±4.62 | -399 (A2C) ±7.76 | -357 (SAC) |
| Inverse Pendulum | Trace 1 | 18.4 ±2.54 | 18.7 ±2.85 | **20.8** **±1.72** | 14.7 ±3.48 | 13.6 (PT-DQN) ±2.27 | 25.7 (TRPO) |
| | Trace 2 | 4.69 ±0.66 | **5.31** **±0.54** | 4.91 ±0.63 | 4.61 ±0.77 | 2.36 (A2C) ±0.45 | 8.51 (TRPO) |
| | Trace 3 | 15.4 ±0.23 | 15.4 ±0.22 | 15.4 ±0.22 | 15.4 ±0.23 | 15.0 (A2C) ±0.46 | 16.5 (TRPO) |
| | Trace 4 | 92.3 ±0.22 | 92.4 ±0.19 | 92.4 ±0.19 | 92.4 ±0.19 | **93.7** (A2C) **±0.21** | 112 (TRPO) |
| Inverse Double Pendulum | Trace 1 | 117 ±5.30 | 114 ±3.54 | 117 ±5.29 | 118 ±5.26 | 118 (A2C) ±9.47 | 165 (TRPO) |
| | Trace 2 | 54.6 ±2.19 | 56.2 ±2.28 | 54.7 ±2.19 | 54.9 ±2.23 | 54.3 (A2C) ±1.72 | 64.0 (TRPO) |
| | Trace 3 | 64.9 ±0.90 | 65.1 ±0.98 | 65.0 ±0.92 | 64.9 ±0.91 | 63.4 (A2C) ±2.37 | 74.0 (TRPO) |
| | Trace 4 | 89.4 ±0.24 | 89.2 ±0.33 | 89.4 ±0.24 | 89.4 ±0.24 | **93.9** (DDQN) **±0.18** | 96.9 (TRPO) |
| Hopper | Trace 1 | 238 ±4.45 | **244** **±5.58** | 241 ±5.37 | 233 ±3.22 | 203 (A2C) ±7.73 | 240 (A2C) |
| | Trace 2 | 141 ±4.13 | **144** **±3.44** | 139 ±3.76 | 135 ±2.54 | 97.9 (A2C) ±3.07 | 131 (TRPO) |
| | Trace 3 | **509** **±2.48** | 508 ±2.54 | 509 ±2.42 | 508 ±2.46 | 484 (A2C) ±13.8 | 520 (A2C) |
| | Trace 4 | **296** **±2.48** | 296 ±2.59 | 291 ±4.23 | 279 ±5.53 | 263 (A2C) ±5.59 | 319 (A2C) |
| Reacher | Trace 1 | -26.6 ±0.15 | -26.5 ±0.21 | -26.5 ±0.19 | -26.4 ±0.26 | **-7.84** (DDQN) **±0.03** | -7.59 (DDQN) |
| | Trace 2 | -19.8 ±0.12 | -19.8 ±0.11 | -19.8 ±0.12 | -19.8 ±0.12 | **-7.78** (DDQN) **±0.07** | -7.41 (DDQN) |
| | Trace 3 | -27.8 ±0.84 | -27.3 ±0.89 | -27.8 ±0.97 | -28.1 ±0.53 | **-8.56** (DDQN) **±0.04** | -8.41 (DDQN) |
| | Trace 4 | -29.5 ±0.31 | -29.4 ±0.29 | -29.4 ±0.73 | -29.3 ±0.83 | **-9.17** (BFDQN) **±0.45** | -7.97 (DDQN) |

Table 9: Average and 95% confidence interval for training time per environment interactions, in $\mu$sec.

| LCPO | A2C | TRPO | DDQN | SAC | Sliding OGD | CLEAR | BFDQN | MBPO | MBCD | Online EWC | PT-DQN |
|---|---|---|---|---|---|---|---|---|---|---|---|
| 0.78 ±0.03 | 0.52 ±0.03 | 0.45 ±0.03 | 0.45 ±0.03 | 0.50 ±0.02 | 0.65 ±0.01 | 0.65 ±0.01 | 0.43 ±0.01 | 0.31 ±0.01 | 2.19 ±0.04 | 5.62 ±0.32 | 3.97 ±0.02 | 0.42 ±0.01 |

online EWC is using SAC, and copied SAC-specific parameters directly). Several LCPO hyperparameters were copied from TRPO, SAC and A2C (namely, entropy target, entropy learning rate, damping coefficient, rollout length, $\lambda$, $\gamma$) and the rest ($c_{anchor}$, $c_{recent}$ and base entropy) were tuned with an informal search with a separate context trace (not in the evaluation set) in Pendulum-v1. The OOD threshold $\sigma$ was not tuned with a search.

Table 10: Average episodic returns and 95th percentile confidence ranges for different algorithms in the Pendulum-v1 environment with discretized and continuous action space.

| Episodic Return | A2C | TRPO | DDQN | SAC | MBPO |
|---|---|---|---|---|---|
| Discrete | -165+-5 | -166+-8 | -149+-2 | -146+-2 | -161+-3 |
| SB3 (Raffin et al., 2021) + RL-Zoo (Raffin, 2020) | -203 | -224 | — | -176 | — |

Table 11: Training setup and hyperparameters for gymnasium environments with external wind.

| Group | Hyperparameter | Value |
|---|---|---|
| Neural network | Hidden layers | (64, 64) |
| | Hidden layer activation | Relu |
| | Output layer activation | Actors: Softmax, Critics and DDQN: Identity mapping |
| | Optimizer | Adam ($\beta_1 = 0.9$, $\beta_2 = 0.999$) (Kingma & Ba, 2017) |
| | Learning rate | Actor: 0.0004, Critic and DDQN: 0.001 |
| | Weight decay | $10^{-4}$ |
| RL training (general) | Random seeds | 25 in main experiments (§5.1), except for MBPO and MBCD 5 seeds in ablations (§5.2, §5.3, §E.1, §D.5, §D.1) |
| | $\lambda$ (for GAE in A2C and TRPO) | 0.9 |
| | $\gamma$ | 0.99 |
| A2C | Rollout per epoch | 200 |
| TRPO | Rollout per epoch | 3200 |
| | Damping coefficient | 0.1 |
| | Stepsize | 0.01 |
| DDQN | Rollout per epoch | 200 |
| | Batch Size | 512 |
| | Initial fully random period | 1000 epochs |
| | $\epsilon$-greedy schedule | 1 to 0 in 5000 epochs |
| | Polyak $\alpha$ | 0.01 |
| | Buffer size $N$ | All samples ($N = 20M$ or $N = 8M$) |
| SAC | Rollout per epoch | 200 |
| | Batch Size | 512 |
| | Initial fully random period | 1000 epochs |
| | Base Entropy | 0.1 |
| | Entropy Target | $0.1\ln(15)$ |
| | Log-Entropy Learning Rate | 1e-3 |
| | Polyak $\alpha$ | 0.01 |
| | Buffer size $N$ | All samples ($N = 20M$ or $N = 8M$) |
| LCPO | Rollout per epoch | 200 |
| | Base Entropy | 0.03 |
| | Entropy Target | $0.1\ln(15)$ |
| | Log-Entropy Learning Rate | 1e-3 |
| | Buffer Size $n_b$ | 1% of samples ($200K$ or $80K$) |
| | Damping coefficient | 0.1 |
| | $c_{anchor}$ | 0.0001 |
| | $c_{recent}$ | 0.1 |
| | $\sigma$ | 1 |
| LCPO-P | PPO Clipping $\epsilon$ | 0.2 |
| | PPO Iterations (Max) | 30 |
| | PPO Max KL | 0.01 |
| | $\kappa$ | 10 |

| Group | Hyperparameter | Value |
|---|---|---|
| MBCD | h | 1000 (default was 100/300) |
| | max_std | 3 (default was 0.5) |
| | N (ensemble size) | 5 |
| | NN hidden layers | (64, 64, 64) |
| MBPO | M (model rollouts) | 512 |
| | N (ensemble size) | 5 |
| | k (rollout length) | 1 |
| | G (gradient steps) | 1 |
| Online EWC | averaging weight $\beta$ | 0.00007 (equivalent to $\sim 3M$ samples at rollout=200) |
| | scaling factor $\alpha$ | 0.1 |
| Sliding OGD | learning rate $\alpha$ | 1e-4 |
| | N (window size) | 1000 episodes |
| CLEAR | Value Clone Coefficient | 1e-2 |
| | Policy Clone Coefficient | 1e-3 |
| | Entropy Coefficient | 5e-3 |
| | V-Trace $\rho$ | 1 |
| | V-Trace c | 1 |
| BFDQN | Depth | 13 |
| | Benna Fusi Buffer Length | 2000 |
| | $g_{1,2}$ | 1e-3 |
| PT-DQN | Permanent Learning Rate | 1e-5 |
| | Transient Learning Rate | 1e-3 |
| | Target Update Period $N$ | 2000 steps |
| | Permanent Update period $K$ | 20000 steps |
| | Transient Forget Factor $\lambda$ | 0.999 |

# APPENDIX G    STRAGGLER MITIGATION

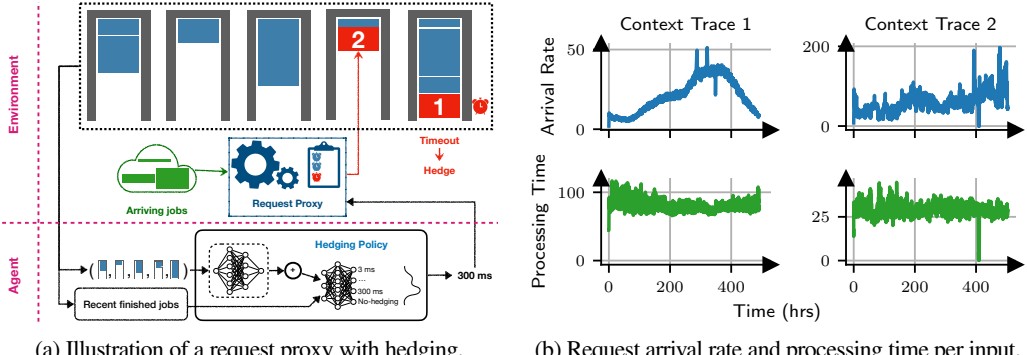

(a) Illustration of a request proxy with hedging.    (b) Request arrival rate and processing time per input.

Figure 8: Request arrival rate and processing time per input.

## G.1    FULL RESULTS

Figure 9 plots the tail latency across experiment time in the straggler mitigation environment for both contexts.

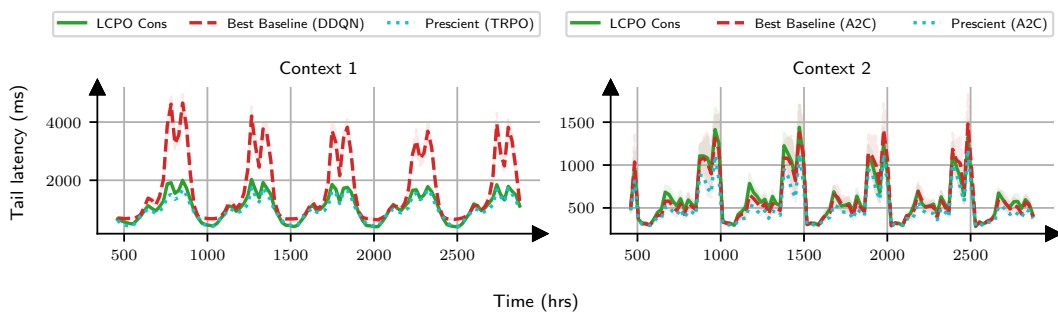

Figure 9: Tail latency with 95th percentile confidence intervals as training progresses (lower is better). We consider an initial learning period of 3.5 million samples. LCPO remains close to the prescient throughout contexts, while baselines suffer from non-stationarity.

## G.2 EXPERIMENT SETUP

We use the straggler mitigation environment from prior work (Hamadanian et al., 2022), with a similar configuration except with 9 actions (timeouts of $600^{ms}$ and $1000^{ms}$ added). Similar to §F.4, our implementations of baselines and LCPO use the Pytorch (Paszke et al., 2019) (v1.13.1, BSD-style license) library. The environment code and dataset is not public and was released to us with a proprietary license. Table 12 is a comprehensive list of all hyperparameters used in training and the environment.

All baselines were tuned on a separate workload. LCPO hyperparameters were copied from the gymnasium experiments, except for base entropy which was tuned with an informal search with a separate workload (not in the evaluation set).

Table 12: Training setup and hyperparameters for straggler mitigation experiments.

| Group | Hyperparameter | Value |
|---|---|---|
| Neural network | Hidden layers | $\phi$ network: (32, 16) |
| | | $\rho$ network: (32, 32) |
| | Hidden layer activation function | Relu |
| | Output layer activation function | Actors: Softmax, Critics and DDQN: Identity mapping |
| | Optimizer | Adam ($\beta_1 = 0.9$, $\beta_2 = 0.999$) (Kingma & Ba, 2017) |
| | Learning rate | 0.001 |
| | Weight decay | $10^{-4}$ |
| RL training (general) | Random seeds | 10 |
| | $\lambda$ (for GAE in A2C and TRPO) | 0.95 |
| | $\gamma$ | 0.9 |
| A2C | Rollout per epoch | 4608 |
| TRPO | Rollout per epoch | 10240 |
| | Damping coefficient | 0.1 |
| | Stepsize | 0.01 |
| DDQN | Rollout per epoch | 128 |
| | Batch Size | 512 |
| | Initial fully random period | 1000 epochs |
| | $\epsilon$-greedy schedule | 1 to 0 in 5000 epochs |
| | Polyak $\alpha$ | 0.01 |
| | Buffer size $N$ | All samples ($N = 21M$) |

| Group | Hyperparameter | Value |
|---|---|---|
| SAC | Rollout per epoch | 128 |
| | Batch Size | 512 |
| | Initial fully random period | 1000 epochs |
| | Base Entropy | 0.01 |
| | Entropy Target | $0.1\ln(9)$ |
| | Log-Entropy Learning Rate | 1e-3 |
| | Polyak $\alpha$ | 0.005 |
| | Buffer size $N$ | All samples ($N = 21M$) |
| LCPO | Rollout per epoch | 128 |
| | Base Entropy | 0.01 |
| | Entropy Target | $0.1\ln(9)$ |
| | Log-Entropy Learning Rate | 1e-3 |
| | Buffer Size $n_b$ | $210K$ |
| | Damping coefficient | 0.1 |
| | $c_{anchor}$ | 0.0001 |
| | $c_{recent}$ | 0.1 |
| MBCD | h | 300000 (default was 100/300) |
| | max_std | 3 (default was 0.5) |
| | N (ensemble size) | 5 |
| | NN hidden layers | (64, 64, 64) |
| MBPO | M (model rollouts) | 512 |
| | N (ensemble size) | 5 |
| | k (rollout length) | 1 |
| | G (gradient steps) | 1 |
| Online EWC | averaging weight $\beta$ | 0.00007 (equivalent to $\sim 2M$ samples at rollout=128) |
| | scaling factor $\alpha$ | 0.1 |

