# OpenReview forum: "Online Reinforcement Learning in Non-Stationary Context-Driven Environments"
_ICLR.cc/2025/Conference — ICLR 2025 Spotlight_

### Official Review · Reviewer_QiEy · 2024-10-28

**Soundness:** 2
**Presentation:** 2
**Contribution:** 2
**Rating:** 5
**Confidence:** 3

**Summary:**

This paper investigates online reinforcement learning (RL) in non-stationary environments,
where the non-stationarity arises from a time-varying exogenous context process affecting the dynamics.
In particular, they study the catastrophic forgetting (CF) problem.
They propose a new method named locally constrained policy optimization (LCPO),
which combats CF by constraining policy updates for a particular context on prior data from other contexts.
They evaluate their method in the classic control tasks from MuJoCo and computer systems environments.

**Strengths:**

Whilst this paper is slightly out of my expertise, I think the authors are addressing an important problem as
handling non-stationarity in online RL is important.
To the best of my knowledge, the method presented in this paper seems original.

I particularly liked Figure 1. I would suggest the authors move it earlier in the paper.

**Weaknesses:**

Overall, I found the paper quite hard to follow.
The method seems fairly straightforward but the paper lacks a clear and easy-to-follow structure I would expect from an ICLR paper.
I am confused as to why Section 4 "Locally-Constrained Policy Optimization" and Section 4 "Methodology" are different sections.
I would recommend the authors combine these sections as they both explain the method.
If needed, use sub-sections to structure the content.

Further to this, the figures are not explained thoroughly enough in the text. What does the y-axis in Figures 3 and 4 represent? I assume it is the cumulative distribution function but this should be stated in the paper.
Further to this, how should the reader interpret the results?
The authors should explain how to interpret the plots, including what the steepness of curves represents, what does it mean
when curves are intersecting, etc?
I suggest the authors check out [rliable](https://github.com/google-research/rliable/tree/master) and use this to create figures.
I am also not sure at what environment step these figures were created for.
Again, this should be clear from the text.

Why are there confidence intervals for Figure 3a but not for Figures 3b or 4?
In Section 6.3, the authors claim that buffer sizes less than $n_b=500$ results in a drop in performance.
I am not sure you can make this claim without showing confidence intervals in Figure 4.

The authors have incorrectly cited throughout the paper.
The authors should take time to become familiar with when to use textual citations and when to use parenthetical citations.
If using the `natbib` package, the authors can use `\citet` for textual citations and `\citep` for parenthetical citations.
The authors almost always cite such that they should be using parenthetical citations but they use textual citations throughout.

What does the bolding in the table represent?
Does it represent statistical significance under a t-test?
This should be clear from the caption.

# Minor comments
- "Mujoco" should be "MuJoCo"
- Line 191 - the policy is stochastic but on line 112 it's a deterministic mapping.

**Questions:**

- How can you improve the clarity of the writing?
- How am I supposed to read Figure 3. What does the y-axis represent?
- Why have you used textual citations everywhere?
- What does the bolding in tables represent?

---

> ### Author Response · Authors · 2024-11-14
>
> We thank the reviewer for their time and consideration. We appreciate the feedback, and will gladly upload a revision of the paper before the discussion deadline that incorporates them.
>
> ---
>
> > Overall, I found the paper quite hard to follow. The method seems fairly straightforward but the paper lacks a clear and easy-to-follow structure I would expect from an ICLR paper. I am confused as to why Section 4 "Locally-Constrained Policy Optimization" and Section 4 "Methodology" are different sections. I would recommend the authors combine these sections as they both explain the method. If needed, use sub-sections to structure the content.
>
> We apologize for the confusion. Section 4 “Locally-Constrained Policy Optimization” is meant to give intuition for why the LCPO constraint would reduce catastrophic forgetting, by comparing the idea to a tabular policy in a discrete space MDP. Section 5 “Methodology” formalizes LCPO, and describes one approach to impose the constraint in practice.
>
> We can combine these two sections to one section with two subsections. We will revise the starting and ending paragraphs of these two sub-sections to improve the overall flow.
>
> ---
>
> > Further to this, the figures are not explained thoroughly enough in the text. What does the y-axis in Figures 3 and 4 represent? I assume it is the cumulative distribution function but this should be stated in the paper. Further to this, how should the reader interpret the results? The authors should explain how to interpret the plots, including what the steepness of curves represents, what does it mean when curves are intersecting, etc? I suggest the authors check out rliable and use this to create figures. I am also not sure at what environment step these figures were created for. Again, this should be clear from the text.
>
> We apologize for the confusion. We hope the explanations below help with clarification.
> * The y-axis in Figures 3 and 4 denotes the CDF, per the label on the y-axis “CDF (%)”. We will amend the captions to also include mentions to the y-axis being CDFs.
> * As for interpretation, we shall revise the main text in line 424 to mention that we seek higher lifelong returns, and will therefore rate baselines by how close their normalized lifelong returns are to 1 across various environments and contexts.
> * As covered in Section 2 (line 116), Lifelong return is the average episodic return across the context trace (after warm-up). The lifelong returns plotted in Figures 3 and 4 are after the entire context trace has finished (lengths are 20M and 8M steps per line 406), and do not pertain to returns of a specific environment step.
> * We will look into rliable and investigate how we can improve our figures using this library. We are thankful for the suggestion.
>
> ---
>
> > Why are there confidence intervals for Figure 3a but not for Figures 3b or 4? In Section 6.3, the authors claim that buffer sizes less than $n_b=500$ results in a drop in performance. I am not sure you can make this claim without showing confidence intervals in Figure 4.
>
> We will be glad to do so, and we can run experiments on more seeds to reduce the variance in confidence intervals if needed. We have reported all confidence intervals in the Appendix (Table 6 for Figure 3b, and Table 7 for Figure 4).
>
> As for why we did not, the point of these experiments was to demonstrate that LCPO is resistant to changes in buffer size and OOD threshold (with the exception of $n_b=25$); most agents in the ablations are very close in performance (e.g., LCPO with $n_b=200K$ vs. LCPO with $n_b=500$), and their confidence intervals will naturally collide. We considered only plotting the confidence interval for $n_b=25$, since we wanted to point out the loss in performance, but ultimately decided against it since we were worried that singling out a specific agent’s confidence interval would cause confusion.
>
> ---
>
> > The authors have incorrectly cited throughout the paper. The authors should take time to become familiar with when to use textual citations and when to use parenthetical citations. If using the natbib package, the authors can use \citet for textual citations and \citep for parenthetical citations. The authors almost always cite such that they should be using parenthetical citations but they use textual citations throughout.
>
> We apologize for this mistake. We used `\cite`, which defaults to `\citep` for numerical citations. We were not aware that for author-year citations, it instead defaults to `\citet`. We will correct citations to use `\citep` throughput the paper.
>
> ---
>
> > What does the bolding in the table represent? Does it represent statistical significance under a t-test? This should be clear from the caption.
>
> We apologize for the confusion, and will amend the captions to explicitly state the bolding. The bolding in Table 5 represents statistical significance under t-test with 25 seeds, and in other tables with less seeds it represents the best performing agent.

---

> > ### Comment · Reviewer_QiEy · 2024-11-26
> >
> > Thank you for your response and addressing my questions.
> >
> > Please can I ask the authors to update the paper with the mentioned changes? I am happy to consider increasing my score but as the ICLR rebuttal allows you to update the paper, I would like to see the changes first. It would also be great if you could colour the new text so it's easy to see.

---

> > > ### Author Response · Authors · 2024-11-26
> > >
> > > We thank the reviewer for their suggestions.
> > >
> > > We have uploaded a revised submission with changed highlighted in red.
> > >
> > > We have added confidence intervals in Figures 3b and 4, and have increased the number of seeds for ablation experiments. However, the full set of seeds have not finished for Figure 3b, and will likely not finish by the deadline. We will update these results with the full set once they are finished.

---

> > > > ### Author Response · Authors · 2024-12-01
> > > >
> > > > We thank the reviewer for their time and insight.
> > > >
> > > > If there are any other comments or feedback, we would be happy to receive and reflect them (unfortunately not able to update the paper in this discussion phase anymore, as the revision deadline has passed).

---

> > > > > ### Comment · Reviewer_QiEy · 2024-12-03
> > > > >
> > > > > I have taken the time to read all reviews, responses, and paper updates. I thank the authors for addressing my questions and updating the paper. I will update my score accordingly.

---

> > > > > > ### Author Response · Authors · 2024-12-03
> > > > > >
> > > > > > We thank the reviewer for their time, feedback and recognition.

---

### Official Review · Reviewer_dCoM · 2024-11-04

**Soundness:** 3
**Presentation:** 3
**Contribution:** 3
**Rating:** 8
**Confidence:** 3

**Summary:**

The paper introduces an online RL method for environments with non-stationary (but observable) context, which affects the dynamics/reward. The main difficulty in such settings is catastrophic forgetting; while training the agent on an unobserved context, it forgets the optimal behavior in past contexts. The authors aim to solve this issue by constraining the agent to maintain current behavior for previously-observed contexts, which is possible due to an OOD detector -- basically a distance measure over contexts. To test the proposed method, the authors use a single toy problem and a large amount of benchmarks, the results show that the method indeed surpasses other baselines.

**Strengths:**

I think the paper is well-written, easy to understand and comprehensive. Notably, the empirical sections present the advantages of the proposed algorithm. For my opinion, with a few minor changes, it passes the acceptance threshold.

### Experiments
The experiments are extensive in terms of both benchmarks and baselines, and show good performance for the proposed method.

### Illustrative Example:
I think this empirical analysis helps to understand what's happening, and shows the value of the method.

**Weaknesses:**

**Paper exceeds the 10-page limit**

### Presentation
1. The references within the paper seem to have improper notation (using § instead of Sec.)

2. I am not sure about the contribution of the opening sentence in the introduction, as this is not the first paper introducing CF, and I do not see any specific relation to the proposed method.

3. Since the method uses a few components, adding a block diagram could help, but is not mandatory in the main body. Such block diagram should show the interconnections at each time step between the buffer, new sampled data, OOD detector, and the agent (policy).

### Related Work:
While not exactly the same, maybe contextual MDPs could be related.

### Minor comments:

1. line 66: "These assumptions are rarely met and lead to poor performance in practice." -- is there any reference to support this claim? If it cannot be backed, you can alter the sentence to be more soft: "...and would likely lead to...".

2. line 268: "proof in §B in the Appendix." -- rephrase.

**Questions:**

1. Have you tried learning the OOD proxy? e.g., using weighted norm with tunable parameters?

2. You do discuss the differences between your proposed OOD detector and CPD, but have you tried comparing the "whole package"? i.e. using LCPO with CPD.

---

> ### Author Response · Authors · 2024-11-14
>
> We thank the reviewer for their time and feedback. We will gladly revise the submission to incorporate their comments.
>
> ---
>
> > Paper exceeds the 10-page limit
>
> If the reviewer is referencing the “Reproducibility Statement”, this does not count towards the page limit (https://iclr.cc/Conferences/2025/AuthorGuide):
> > “This optional reproducibility statement will not count toward the page limit, but should not be more than 1 page.”
>
> ---
>
> > The references within the paper seem to have improper notation (using § instead of Sec.)
>
> We apologize for the confusion. The § symbol is called the “section” symbol, and is used instead of “Section” in papers, including published ICLR papers [1]. Has ICLR required submissions to not use the symbol this year? We’d be glad to correct the text if this is the case.
>
> [1] Kong, X., Huang, W., & Liu, Y. Conditional Antibody Design as 3D Equivariant Graph Translation. In The Eleventh International Conference on Learning Representations.
>
> ---
>
> > I am not sure about the contribution of the opening sentence in the introduction, as this is not the first paper introducing CF, and I do not see any specific relation to the proposed method.
>
> We apologize for the confusion. To be clear, we do not claim to the be first paper studying CF. The relevance of the opening sentence is hinting at the importance of not forgetting the past, and LCPO operates by constraining the agent to not forget the past.
>
> ---
>
> > Since the method uses a few components, adding a block diagram could help, but is not mandatory in the main body. Such block diagram should show the interconnections at each time step between the buffer, new sampled data, OOD detector, and the agent (policy).
>
> We thank the reviewer for the suggestion, and will gladly create a block diagram for LCPO.
>
> ---
>
> > While not exactly the same, maybe contextual MDPs could be related.
>
> Thank you for the suggestion. Could the reviewer kindly clarify if they are referring to “Contextual Markovian Decision Processes” [2], or broadly papers that focus on contextual MDPs? The works mentioned in lines 142-151 are focused on context-driven MDPs, and we will also add [2] to this section.
>
> [2] Hallak, A., Di Castro, D., & Mannor, S. (2015). Contextual markov decision processes. arXiv preprint arXiv:1502.02259.
>
> ---
>
> > Have you tried learning the OOD proxy? e.g., using weighted norm with tunable parameters?
>
> This sounds interesting, but could the reviewer kindly clarify their suggestion?
>
> We briefly experimented with auto-encoders + clustering algorithms operating on the latent space, but ultimately found that simpler OOD metrics worked well enough. We also experimented with online gaussian mixture models but found them to be unstable on context traces from real workloads (line 409 and Figure 7b).
>
> ---
>
> > You do discuss the differences between your proposed OOD detector and CPD, but have you tried comparing the "whole package"? i.e. using LCPO with CPD.
>
> We briefly experimented mixing MBCD with LCPO on Pendulum-v1. The results were not promising. Upon further investigation, we found the reported change-points to be noisy, similar to that shown in Figure 2. We did not include these experiments in the paper.
>
> Despite this, we believe it is too soon to rule out CPD+LCPO; a “soft” version of CPD algorithms could ultimately benefit LCPO. MBCD is trying to find change-points in context traces that do not have any, and will understandably find it difficult. But if we relax the definition of change-points from hard binary shifts to fuzzy continuous shifts, it may circumvent the issues we found. We found exploring this avenue to be outside the scope of this paper, but an interesting future direction.
>
> ---
>
> > 1. line 66: "These assumptions are rarely met and lead to poor performance in practice." -- is there any reference to support this claim? If it cannot be backed, you can alter the sentence to be more soft: "...and would likely lead to...".
>
> The comment in line 66 is from our own analysis (D.1, D.2 and D.3 in Appendix, ) and empirical evaluation (Section 6.1 and D.1 in the Appendix). We will revise the text to be softer, and will include references to these sections for clarity.

---

> > ### Comment · Reviewer_dCoM · 2024-11-24
> >
> > Thank you for addressing my concerns.
> > See my comments below. For now, I tend to leave my original rating, although the other reviewers responses may change that.
> >
> > 1. Regarding the length of the paper: you are correct.
> >
> > 2. The symbol § for section ref: I am not certain.
> >
> > 3. The opening sentence: I am neutral, although I think the community is well-aware of the CF issue.
> >
> > 4. Could you upload your suggested block diagram?
> >
> > 5. Contextual MDPs - I think the papers that define contextual MDPs are enough here, as the problem setting is different, which should be emphasized in the paper.
> >
> > 6. Learning the OOD proxy: for example, learn a metric over the contextual space that aims to maximize the "long term" performance.

---

> ### Author Response · Authors · 2024-11-26
>
> We thank the reviewer for their responses.
>
> ---
>
> > Could you upload your suggested block diagram?
>
> We have added an overview of LCPO's overall training pipeline in Appendix C, page 20, Figure 5, in the revised submission. We appreciate any feedbacks around this figure.
>
> ---
>
> > Learning the OOD proxy: for example, learn a metric over the contextual space that aims to maximize the "long term" performance.
>
> We have not attempted to learn a metric over the contextual space, though the idea is interesting. There might be ways to tie long-term performance to metric definitions directly through meta-learning.

---

> > ### Comment · Reviewer_dCoM · 2024-11-27
> >
> > Regarding the diagram, I think it only lacks cosmetics; rounded corners for the arrows, add some color etc.
> >
> > As you addressed my concerns, I decided to raise my score.

---

> > > ### Author Response · Authors · 2024-12-03
> > >
> > > We thank the reviewer for their recognition of our work. We will apply these suggestions to the diagram.

---

### Official Review · Reviewer_ksTz · 2024-11-04

**Soundness:** 4
**Presentation:** 4
**Contribution:** 3
**Rating:** 8
**Confidence:** 4

**Summary:**

The paper addresses "catastrophic forgetting" in continual reinforcement learning with non-stationary dynamics. The dynamics depends on the current context which is observable but can change arbitrarily. A problem in these kind of settings is that the agent over time forgets how to behave in prior contexts. To address this problem the paper proposes a new on-policy algorithm that aims to keep policy updates such that the policy for prior contexts does not change. The proposed approach does not need task labels but only an out-of-distribution detector.

**Strengths:**

Well written paper. Clear definitions and derivations. The paper explains well all the design decisions in the proposed approach.

The main algorithmic idea of sampling states from the history buffer that differ from the samples currently collected from the environment, and, then using these samples to constrain the policy update is simple and based on the experimental results appears very effective.

I like that the resulting algorithm is an almost straight forward modification of TRPO but for a completely different problem. Note that while being simple the main idea is only obvious in hindsight.

**Weaknesses:**

I could not find major weaknesses in the paper.

This is not a weakness but just a comment:
Choosing of the out of distribution (OOD) detector is one obvious design decision when running the method in a specific environment. Section 6.2 discusses different thresholds for the OOD detector. Although OOD detection has been discussed in the supervised learning literature, the paper could provide a more thorough discussion on which kind of OOD detectors could be beneficial in which kind of RL environments.

**Questions:**

The proposed algorithm places a constraint that aims to keep the new policy relevant for old samples. This constraint replaces the TRPO policy update constraint. The original TRPO formulation results in an approximate natural gradient due to which in TRPO the gradient update does not in principle depend on the parameterization of the (neural network) policy. This property is lost here? Do we get other properties?

---

> ### Author Response · Authors · 2024-11-14
>
> We thank the reviewer for their time and feedback. We will incorporate their comments in a revision of the paper.
>
> ---
>
> > This is not a weakness but just a comment: Choosing of the out of distribution (OOD) detector is one obvious design decision when running the method in a specific environment. Section 6.2 discusses different thresholds for the OOD detector. Although OOD detection has been discussed in the supervised learning literature, the paper could provide a more thorough discussion on which kind of OOD detectors could be beneficial in which kind of RL environments.
>
> This is a great suggestion. We can write a section in the Appendix that discusses what the OOD function aims to do, what class of OOD functions exist in the literature, and how to match different context dynamics to different OOD classes.
>
> ---
>
> > The proposed algorithm places a constraint that aims to keep the new policy relevant for old samples. This constraint replaces the TRPO policy update constraint. The original TRPO formulation results in an approximate natural gradient due to which in TRPO the gradient update does not in principle depend on the parameterization of the (neural network) policy. This property is lost here? Do we get other properties?
>
> Unfortunately the property is lost. LCPO’s constraint is being applied in a different part of the policy space (i.e., policy conditioned on OOD states) then the part the optimization is happening in (i.e., policy conditioned on recent states), and the parametrization of the policy becomes important in defining how these two different pieces of the policy distribution are related.
>
> Formally, the LCPO distance metric itself can be approximated with $(\theta-\theta_0)^T.\hat{F}.(\theta-\theta_0)$, where $\hat{F}$ is the Fisher Information Matrix w.r.t to the OOD state-context $<\hat{s}, \hat{z}>$. This distance metric is invariant w.r.t to the parametrization of the policy on the OOD state-context $\pi(\cdot|\hat{s}, \hat{z})$. However the optimization objective is w.r.t to some *recent* state-context $<s, z>$. The distance defined by $\hat{F}$ is not invariant w.r.t the parametrization of the policy for the recent state-context $\pi(\cdot|s, z)$. They are different points in the policy manifold, and without some underlying parameterization to tie them together, the constraint does not affect the objective.

---

> > ### Comment · Reviewer_ksTz · 2024-11-25
> >
> > Thanks to the authors for the response. I will keep my score.

---

> > > ### Author Response · Authors · 2024-11-26
> > >
> > > We thank the reviewer for their comments and recognition of our work.

---

### Official Review · Reviewer_JEGP · 2024-11-07

**Soundness:** 3
**Presentation:** 3
**Contribution:** 3
**Rating:** 8
**Confidence:** 4

**Summary:**

The paper proposes Locally Constrained Policy Optimization (LCPO), a policy gradient algorithm for non-stationary reinforcement learning where the context changes are observed and exogenous. LCPO’s updates are constrained to preserve the policy trained on the past contexts while it maximizes returns on newer samples. Two replay buffers are maintained in order to store the samples belonging to the past contexts and the current context. Whenever the distribution of samples is different in these two buffers, which is detected using an OOD threshold-based detector, then a “regularization” update is performed to preserve past learnings. The proposed approach is tested on a range of environments with a whole suite of baselines to demonstrate its benefits.

**Strengths:**

* The paper is well-written and easy to read: the introduction is thorough, the related works section is exhaustive, preliminary and the LCPO sections provide adequate details, and the methodology section is well-presented.
* The specific kind of non-stationary problem that the paper considers is well-described. And, the proposed solution fits like a glove to this problem. The approach is also intuitive and parallels the TRPO algorithm.
* The authors evaluate their approach against numerous baselines on several tasks, which provides substantial confidence in the presented results and conclusions. The ablation studies are also welcome, as they offer more insights into the proposed approach.
* The illustrative examples discussed throughout the paper offer an intuitive understanding of various complex ideas – the authors have done a great job of including them.

**Weaknesses:**

* The biggest weakness is that the paper considers a specific kind of non-stationarity in the study. Although exogenous processes could be common in real-world scenarios, many other kinds of non-stationarities exist and it is unclear whether the proposed approach works well in those cases. Besides, the OOD detector is an important piece in the algorithm and its effectiveness depends on the threshold value – a hyperparameter that is environment-dependent and unknown beforehand.
* The approach also assumes that the context is observed, although it is not used as a task detector or task boundary detector. In many practical cases, the context information is unavailable and it is unclear how one could use the proposed approach in that case.
* The proposed constrained optimization problem is approximately solved, which in itself is fine, but when paired with the fact that the approach doesn’t use TRPO-style local constraint, the training procedure could be unstable.

**Questions:**

**Decision:**

Although the paper has some weaknesses, the positives outweigh them. The paper is well-executed and it is a good contribution to continual reinforcement learning. Therefore, I recommend an **acceptance.**

**Areas of improvement:**

* The paper should discuss how [1] and [2] are related to the proposed approach. [1] uses a KL constraint on the current policy to be closer to the global policy, and [2] learns a global value function as a baseline estimate for all contexts on top of which the learning happens. It seems like the global policy and the global value functions induce the regularization effect on the past that the paper proposes.
* In Sec 4, the episodic returns are -4 and -6 and not -3 and -5 as mentioned in the paper.
The ablation experiments and the results presented in the appendix can be summarized as one-line bullet points in the main paper.

**Questions:**

* Deep neural networks are known to overfit to the early samples [3], wouldn’t regularizing with respect to the past data exacerbate this process?
* How is the context information different from a task ID?
* In Sec 4, does the A2C use both context and state information to learn the policy?
* Is it possible to approximate the proposed objective in the PPO style as opposed to the TRPO style to avoid using second-order information?
* What happens if the reverse KL is used in the constraint proposed in Eq. 1?
* What is a warm-up period and why is it needed in the experiments?

**References:**

[1] Teh, Yee, et al. "Distral: Robust multitask reinforcement learning." Advances in neural information processing systems 30 (2017).

[2] Anand, Nishanth, and Doina Precup. "Prediction and control in continual reinforcement learning." Advances in Neural Information Processing Systems 36 (2024).

[3] Nikishin, Evgenii, et al. "The primacy bias in deep reinforcement learning." International conference on machine learning. PMLR, 2022.

---

> ### Author Response · Authors · 2024-11-14
>
> We thank the reviewer for their time and comments. We will thoroughly revise the paper to reflect their suggestions.
>
> ---
>
> > The biggest weakness is that the paper considers a specific kind of non-stationarity in the study. Although exogenous processes could be common in real-world scenarios, many other kinds of non-stationarities exist and it is unclear whether the proposed approach works well in those cases. Besides, the OOD detector is an important piece in the algorithm and its effectiveness depends on the threshold value – a hyperparameter that is environment-dependent and unknown beforehand.
>
> These are fair points. If the context is non-exogenous, the current formulation of the problem will lead to an adversarial POMDP, which may be in general difficult, if not impossible to learn a policy for.
>
> We view the OOD detector as the single source of domain knowledge from the user. Without any knowledge of the context trace beforehand, it would again be difficult, if not impossible, to reason about how different two pieces of context are. It may be possible if we are allowed to see context traces from the same distribution, akin to the problem setups in Meta RL. This would be an interesting future direction.
>
> ---
>
> > The approach also assumes that the context is observed, although it is not used as a task detector or task boundary detector. In many practical cases, the context information is unavailable and it is unclear how one could use the proposed approach in that case.
>
> We agree with the reviewer that this problem setup does not encompass all possible settings of context-driven MDP. In cases where the context is latent, one could potentially combine LCPO with approaches that infer the latent context, such as those in lines 142-151.
>
> ---
>
> > The proposed constrained optimization problem is approximately solved, which in itself is fine, but when paired with the fact that the approach doesn’t use TRPO-style local constraint, the training procedure could be unstable.
>
> While the TRPO-style constraint is not enforced in the optimization stage, it is enforced in the line search stage (line 364-265, lines 9-10 in Algorithm 1). We do this by reducing the step size until the local TRPO constraint is also met.
>
> ---
>
> > The paper should discuss how [1] and [2] are related to the proposed approach. [1] uses a KL constraint on the current policy to be closer to the global policy, and [2] learns a global value function as a baseline estimate for all contexts on top of which the learning happens. It seems like the global policy and the global value functions induce the regularization effect on the past that the paper proposes.
>
> We thank the reviewer for their suggestions. We will add these papers to the related works section. Below, we discuss their relation to our work:
>
> * Distral [1] is a technique for the multi-task RL problem setup, where the goal is to learn multiple tasks in parallel faster than learning them separately, i.e. the goal here is positive transfer learning. Distral suggests learning a distilled policy that represents the shared dynamics of the multiple tasks. This distilled policy is used to direct task-specific policies towards faster convergence. This problem setup is different than ours and does not focus on continual RL or catastrophic forgetting.
> * PT-Q learning [2] is a technique focused on continual RL. The general idea is to learn a slow-changing Q function $Q^{(P)}(s, a; \theta)$ and combine it with a fast changing & decayed Q function $Q^{(T)}(s, a, \mathbf{w})$ to learn the environment. The slow-changing Q function allows learning a good bootstrap network that allows the fast changing Q function to rapidly converge to the optimal policy. We would gladly add PT-Q learning as a baseline, and will update the paper if the results are finished before the deadline.
>
> ---
>
> > In Sec 4, the episodic returns are -4 and -6 and not -3 and -5 as mentioned in the paper. The ablation experiments and the results presented in the appendix can be summarized as one-line bullet points in the main paper.
>
> We apologize for the confusion. We should clarify in the main text, that once the agent reaches the terminal state, the reward is 0. Thus the last step leading to the terminal state does not accrue a reward of -1.

---

> ### Author Response · Authors · 2024-11-14
>
> > Deep neural networks are known to overfit to the early samples [3], wouldn’t regularizing with respect to the past data exacerbate this process?
>
> It may exacerbate it, if exploration is not encouraged. If the context traces are truly different for past data and the neural network has “enough capacity”, the neural network should be able to learn a new policy for the new data without forgetting the old one. But to avoid overfitting and sub-optimal convergence, we need to motivate the agent to explore in new contexts. In this paper, we do this through entropy regularization (Eq. 3), but it could be done through any exploration methodology, e.g. curiosity-based exploration.
>
> ---
>
> > How is the context information different from a task ID?
>
> In summary, context information is a generalization of task ID. Task IDs are discrete indices and clearly separable, while the context process is continuous, multi-dimensional and can change gradually or abruptly.
>
> A task ID is a discrete and often bounded (2 to ~100 tasks, depending on the problem setup) variable, and for task IDs $i$ and $j$, it ensures that the MDPs subject to task IDs $i$ and $j$ are the same iff $i=j$. Task IDs ensure that the underlying MDP is piece-wise stationary; for any period of time where the task ID remains unchanged, the MDP behaves exactly the same way. When task IDs are available, we can solve CF by numerous ways such as learning separate policies per task, forcing gradient updates that are orthogonal to prior tasks, etc.
>
> None of these are possible with context information. The context process is a continuous variable (possibly multi-dimensional) that may change slowly or quickly across time. Since it is continuous, we cannot learn a separate policy per each context variable anymore (there are millions of different context values for a 1-dimensional variable). We cannot force orthogonal updates since small changes in the context could be considered different “tasks”, but would in practice be too similar to the current “task”.  The clear separation that task IDs promise does not exist in the context information.
>
> Strictly speaking, all task IDs are context variables, but context variables cannot be reduced to task IDs. We can only do so in the unique case where the context process is known to be piece-wise stationary. If we make such an assumption, we can map context values to task IDs, which is what change-point detection methods do. When the context process is not piece-wise stationary, change-point detection leads to noisy task IDs, such as that observed in Figure 2.
>
> Therefore, context information allows us to model and solve problems where the MDP changes may be slow and gradual, where there is no clear separation in MDP behavior across time. They also allow modeling problems where context can be multi-dimensional instead of single dimensional.
>
> ---
>
> > In Sec 4, does the A2C use both context and state information to learn the policy?
>
> Yes. A2C, Tabular A2C and LCPO all observe both the state $s_t$ and context $z_t$.
>
> ---
>
> > Is it possible to approximate the proposed objective in the PPO style as opposed to the TRPO style to avoid using second-order information?
>
> It is approximately possible, but not fully. In E in the Appendix, we experiment with a PPO style agent, where the KL distance is added as an auxiliary loss function (Eq. 18, page 25) to the PPO loss function (Eq. 19, page 25). It does not work as well as LCPO, since this formulation is trying to solve the Lagrangian dual of the original problem, but the Lagrange coefficient ($\kappa$ in Eq. 19) can change for different contexts, making this approach hyperparameter sensitive.
>
> It is not possible to embed the LCPO constraint in the proximal update rule of PPO. The proximal update rule needs the advantage value for whatever state it is being applied to (as PPO is an on-policy approach), and since we do not have advantage values for OOD states visited in past data, we cannot apply the constraint. Note that the advantage values we observed when we were collecting past data is obsolete, since the policy has changed since then, and cannot be used in an on-policy approach.
>
> ---
>
> > What is a warm-up period and why is it needed in the experiments?
>
> The goal of the problem setup is to optimize for lifelong returns—defined in Section 2, line 116—which are the asymptotic average episodic return across time. Lifelong returns cares about the eventual performance of the policy, and will ignore transient returns where the policy is exploring. In our empirical evaluation, context traces are finite in length. Therefore in calculating lifelong returns we exclude a portion of experiment at the start so the lifelong return is not dominated by transient exploration dynamics. This is particularly important since different approaches use different exploration strategies, e.g., DDQN using eps annealing vs. entropy regularization for actor critic agents.

---

> ### Author Response · Authors · 2024-11-14
>
> > What happens if the reverse KL is used in the constraint proposed in Eq. 1?
>
> Empirically, Algorithm 1 would not change. When observing the distributions $\pi(\cdot|s, z, \theta)$ and $\pi(\cdot|s, z, \theta_0)$, the forward and reverse KL are equivalent, at least up to the second order Taylor expansion at $\theta=\theta_0$. We show this below.
>
> The minima for the forward and reverse KL is $\theta=\theta_0$, and therefore the zeroth and first order terms of the Taylor expansion will be zero.
>
> As for the second order, for the forward KL we can derive the second order gradient and show that it is equal to the corresponding Fisher Information Matrix (FIM) entry.
>
> For a shorthand, define $f(a,\\theta):=\\pi(a|s, z, \\theta)$, for some $ s \in \mathcal{S}, z\in \mathcal{Z}$.
>
> We have:
> \begin{align}
>    \frac{\partial^2D_{KL}\left(f(a,\theta_0)||f(a,\theta)\right)}{\partial\theta_i\partial\theta_j}|_{\theta=\theta_0} &= \\left[\frac{\partial^2}{\partial\theta_i\partial\theta_j} \sum\_{a \in \mathcal{A}}\left[f(a,\theta_0)\\left(\ln(f(a,\theta_0))-\ln(f(a,\theta))\right)\right]\right]|\_{\theta=\theta_0}\\\\
>    &=\\left[\frac{\partial}{\partial\theta_j}\sum\_{a\in \mathcal{A}}\left[f(a,\theta_0)\left(-\frac{\partial f(a,\theta)}{\partial\theta_i}\frac{1}{f(a,\theta)}\right)\right]\right]|\_{\theta=\theta_0}\\\\
>    &=\sum\_{a\in\mathcal{A}}\left[f(a,\theta_0)\left(-\frac{\partial^2 f(a,\theta)}{\partial\theta_i\partial\theta_j}\frac{1}{f(a,\theta)}+\frac{\partial f(a,\theta)}{\partial\theta_i}\frac{\partial f(a,\theta)}{\partial\theta_j}\frac{1}{f(a,\theta)^2}\right)\right]\_{\theta=\theta_0}\\\\
>    &=\sum\_{a\in\mathcal{A}}\left[f(a,\theta_0)\left(-\frac{\partial^2 f(a,\theta)}{\partial\theta_i\partial\theta_j}|\_{\theta=\theta_0}\frac{1}{f(a,\theta_0)}+\\left[\frac{\partial f(a,\theta)}{\partial\theta_i}\frac{\partial f(a,\theta)}{\partial\theta_j}\frac{1}{f(a,\theta)^2}\right]|\_{\theta=\theta_0}\right)\right]\\\\
>    &=-\sum\_{a\in\mathcal{A}}\left[\frac{\partial^2 f(a,\theta)}{\partial\theta_i\partial\theta_j}|\_{\theta=\theta_0}\right]+\sum\_{a\in\mathcal{A}}\left[f(a,\theta_0)\frac{\partial \ln f(a,\theta)}{\partial\theta_i}|\_{\theta=\theta_0}\frac{\partial \ln f(a,\theta)}{\partial\theta_j}|\_{\theta=\theta_0}\right]\\\\
>    &=-\frac{\partial^2}{\partial\theta_i\partial\theta_j}\\left[\sum\_{a\in\mathcal{A}}f(a,\theta)\right]|\_{\theta=\theta_0}+\mathbb{E}\_{a\sim f(a, \theta_0)}\\left[\frac{\partial \ln f(a,\theta)}{\partial\theta_i}|\_{\theta=\theta_0}\frac{\partial \ln f(a,\theta)}{\partial\theta_j}|\_{\theta=\theta_0}\right]\\\\
>    &=-\frac{\partial^2}{\partial\theta_i\partial\theta_j}[1]|\_{\theta=\theta_0}+\mathbb{E}\_{a\sim f(a, \theta_0)}\\left[\frac{\partial \ln f(a,\theta)}{\partial\theta_i}|\_{\theta=\theta_0}\frac{\partial \ln f(a,\theta)}{\partial\theta_j}|\_{\theta=\theta_0}\right]\\\\
>    &=G\_{i,j}
> \end{align}
>
> where G is the FIM.
>
> Now for the reverse KL, we can similarly derive the second order gradient:
>
> \begin{align}
>    \frac{\partial^2D_{KL}\left(f(a,\theta)||f(a,\theta_0)\right)}{\partial\theta_i\partial\theta_j}|\_{\theta=\theta_0} &= \left[\frac{\partial^2}{\partial\theta_i\partial\theta_j}\sum\_{a\in\mathcal{A}}f(a,\theta)\left[\ln(f(a,\theta))-\ln(f(a,\theta_0))\right]\right]|\_{\theta=\theta_0}\\\\
>    &=\left[\frac{\partial}{\partial\theta_j}\sum\_{a\in\mathcal{A}}\left(\frac{\partial f(a,\theta)}{\partial\theta_i}\left[\ln(f(a,\theta))-\ln(f(a,\theta_0))\right]+f(a,\theta)\frac{\partial f(a,\theta)}{\partial\theta_i}\frac{1}{f(a,\theta)}\right)\right]|\_{\theta=\theta_0}\\\\
>    &=\sum\_{a\in\mathcal{A}}\left[\frac{\partial^2 f(a,\theta)}{\partial\theta_i\partial\theta_j}\left(\ln(f(a,\theta))-\ln(f(a,\theta_0))\right)+\frac{\partial f(a,\theta)}{\partial\theta_i}\frac{\partial f(a,\theta)}{\partial\theta_j}\frac{1}{f(a,\theta)}+\frac{\partial^2 f(a,\theta)}{\partial\theta_i\partial\theta_j}\right]|\_{\theta=\theta_0}\\\\
>    &=\sum\_{a\in\mathcal{A}}\left[\frac{\partial^2 f(a,\theta)}{\partial\theta_i\partial\theta_j}|\_{\theta=\theta_0}\left(\ln(f(a,\theta_0))-\ln(f(a,\theta_0))\right)\right]+\sum\_{a\in\mathcal{A}}\left[f(a,\theta_0)\left(\frac{\partial f(a,\theta)}{\partial\theta_i}\frac{\partial f(a,\theta)}{\partial\theta_j}\frac{1}{f(a,\theta)^2}\right)\right]_{\theta=\theta_0}+\sum\_{a\in\mathcal{A}}\left[\frac{\partial^2 f(a,\theta)}{\partial\theta_i\partial\theta_j}\right]|\_{\theta=\theta_0}\\\\
>    &=0+\mathbb{E}\_{a\sim f(a,\theta_0)}\left[\frac{\partial \ln f(a,\theta)}{\partial\theta_i}|\_{\theta=\theta_0}\frac{\partial \ln f(a,\theta)}{\partial\theta_j}|\_{\theta=\theta_0}]\right]+0\\\\
>    &=G\_{i,j}
> \end{align}
>
> Since the second order derivatives for the forward and reverse KL at $\theta=\theta_0$ turned out to be the same, the approach in Eq. 4 would be the same. Therefore, if we use reverse KL in Eq. 1, we would reach the same Eq. 4 as we would with forward KL.

---

> > ### Comment · Reviewer_JEGP · 2024-11-26
> >
> > I thank the authors for responding to my review. The explanation provided by the authors in their rebuttal clarifies my questions. I strongly encourage them to include these points of clarification in their draft. Including the PT-DQN baseline in experiments is also appreciated. Given these, I will increase my score to a **clear acceptance**.

---

> ### Author Response · Authors · 2024-11-26
>
> We thank the reviewer for their feedback and for their recognition of our work.
>
> We have included a comparison to PT-DQN in the revised submission. In short, LCPO is better in performance than PT-DQN, but PT-DQN does outperform DDQN in certain environments and traces.

---

### Author Response · Authors · 2024-11-26

We thank the reviewers for their feedback. We have uploaded a revision of the paper, where changes are denoted by red text. We also outline the changes below:
* Comparison to a new baseline, PT-DQN [1].
* Experiments with a larger set of seeds, and figures with confidence intervals for the ablation studies in section 5.2 (sensitivity to OOD metric) and section 5.3 (sensitivity to buffer size).
* Added a block diagram of LCPO's overall training pipeline (Figure 5) in Appendix C.
* Added suggested related work [1, 2, 3].
* Merged the prior section 4 (Locally-Constrained Policy Optimization) and section 5 (Methodology) to one section.
* Corrected citation errors.
* Revised bolding in tables, such that bolding is only applied to statistically significant results, and is clearly defined in the table caption.
* Clarified the definition of episodic reward in section 4.1 (Illustrative example).
* Clarified figures 3a, 4b and 4; mentioned in captions that the Y-axis is a CDF (%), and mentioned in text what the intended outcome is for these figures.
* Corrected inconsistent notations in section 4.2 (Methodology) and Appendix E (LCPO variants).
* Applied suggestions for text in lines 66 and 245.


There are still changes to be made that will likely not be ready by the deadline:
* Section 4 (Locally-Constrained Policy Optimization) needs some revision in text to improve flow.
* We shall write a section in the appendix around what OOD metrics are useful for LCPO, and what varieties are commonly used in OOD research.
* The extra experiments for confidence intervals in Figures 3b did not finish running for the full set of seeds. We shall update Figure 3b and Table 6 when the experiments are finalized.

---

[1] Anand, Nishanth, and Doina Precup. "Prediction and control in continual reinforcement learning." Advances in Neural Information Processing Systems 36 (2024).

[2] Teh, Yee, et al. "Distral: Robust multitask reinforcement learning." Advances in neural information processing systems 30 (2017).

[3] Hallak, A., Di Castro, D., & Mannor, S. (2015). Contextual markov decision processes. arXiv preprint arXiv:1502.02259.

---

### Meta-Review · Area_Chair_dvVG · 2024-12-24

**Metareview:**

The paper proposes Locally Constrained Policy Optimization (LCPO), which is a policy gradient method for changing environments. The reviewers highlight that the problem setting is well specified and reasonable, the proposed algorithm is a straightforward extension of TRPO, and that the illustrative examples are valuable. The primary highlighted weaknesses was around the context availability/information patterns and the OOD threshold method.

Three of the four reviewers recommend acceptance with the last reviewer leaning reject. The last reviewer has primarily raised issues around paper formatting. I recommend acceptance of the paper, but I would ask the authors to make sure they have addressed all of these formatting issues.

**Additional Comments On Reviewer Discussion:**

Several questions around paper formatting were raised (eg table bolding). I _believe_ these have all now been addressed (the authors have stated explicitly that they addressed many formatting issues), based on the discussion.

There was substantial discussion around the context information and task boundary information. Reviewers overall are satisfied with the scope of the paper, and I agree that the paper covers a reasonable problem setting and excluding relaxed versions of the problem (in terms of eg context knowledge) is reasonable.

---

### Decision · Program_Chairs · 2025-01-22

Accept (Spotlight)